

# Synthesizing long-term sea level rise projections - the MAGICC sea level model

Alexander Nauels[1], Malte Meinshausen[1,2], Matthias Mengel[2,3], Katja Lorbacher[1], and Tom M. L. Wigley[4,5]

[1]Australian-German Climate and Energy College, The University of Melbourne, Parkville 3010, Victoria, Australia
[2]Potsdam Institute for Climate Impact Research, Telegrafenberg A26, 14412 Potsdam, Germany
[3]Physics Institute, Potsdam University, 14476 Potsdam, Germany
[4]The Environment Institute and School of Biological Sciences, The University of Adelaide, SA 5005, Australia
[5]Climate and Global Dynamics Division, National Center for Atmospheric Research, Boulder, CO 80307-3000, USA

*Correspondence to:* Alexander Nauels (alexander.nauels@climate-energy-college.org)

**Abstract.** Sea level rise is one of the major impacts of global warming; it will threaten coastal populations, infrastructure, and ecosystems around the globe in coming centuries. Well-constrained sea level projections are needed to estimate future losses from Sea Level Rise (SLR) and benefits of climate protection and adaptation. Process-based models that are designed to resolve the underlying physics of individual sea level drivers form the basis for state-of-the-art sea level projections. However, 5 associated computational costs allow for only a small number of simulations based on selected scenarios that often vary for different sea level components. This approach does not sufficiently support sea level impact science and climate policy advice, which require a sea level projection methodology that is flexible with regard to the climate scenario yet comprehensive and bound to the physical constraints provided by process-based models. To fill this gap, we present a sea level model that emulates global mean long-term process-based model projections for all major sea level components. Thermal expansion estimates are 10 calculated with the hemispheric upwelling-diffusion ocean component of the simple carbon cycle-climate model MAGICC, which has been updated and calibrated against CMIP5 ocean temperature profiles and thermal expansion data. Global glacier contributions are estimated based on a parameterization constrained by transient and equilibrium process-based projections. Sea level contribution estimates for Greenland and Antarctic ice sheets are derived from surface mass balance and solid ice discharge parameterizations reproducing current output from ice-sheet models. The land water storage component replicates 15 the latest hydrological modeling results. For 2100, we project 0.38 m to 0.59 m (66% range) total SLR based on the RCP2.6 scenario, 0.48 m to 0.68 m for RCP4.5, 0.48 m to 0.72 m for RCP6.0, and 0.67 m to 0.97 m for RCP8.5. These projections lie within the range of the latest IPCC SLR estimates. SLR projections for 2300 yield median responses of 0.97 m for RCP2.6, 1.66 m for RCP4.5, 2.32 m for RCP6.0, and 5.12 m for RCP8.5. The MAGICC sea level model provides a powerful and efficient platform for probabilistic uncertainty analyses of long-term SLR projections. It can be used as a tool to directly 20 investigate the SLR implications of different mitigation pathways and may also serve as input for regional SLR assessments via component-wise sea level pattern scaling.





# 1 Introduction

Global sea level has increased by around 0.2 m since the beginning of the 20th century and will continue to rise during the 21st century and far beyond (Church and White, 2011; Church et al., 2013a). This will have wide-ranging impacts for coastal regions around the globe and therefore requires careful monitoring. The total sea level signal is the sum of several individual sea level components, the main ones being thermal expansion, global glacier melt, Greenland and Antarctic ice sheet mass loss and land water storage changes (Church et al., 2013a). Over the coming centuries, the magnitude of total SLR will strongly depend on the amount of anthropogenic Greenhouse Gases (GHG) emitted to the atmosphere during the 21st century and the corresponding physical responses of the major SLR drivers (Horton et al., 2014). Future GHG emissions are therefore a main uncertainty source when trying to project SLR trajectories. SLR uncertainties are further increased by structural differences of the underlying process-based models for the individual SLR contributions and limited process understanding, like the behavior of polar ice shelves in a warming world (Nicholls and Cazenave, 2010). To assess major parts of these scenario and model uncertainties, we extend the widely used simple carbon cycle-climate model MAGICC (Meinshausen et al., 2011a, 2009; Wigley et al., 2009; Wigley and Raper, 2001) to comprehensively model global sea level rise. This MAGICC sea level model has been designed to emulate the behavior of the latest available process-based sea level projections, with thorough calibrations for each major sea level component. It is intended to serve as an efficient and flexible tool for the assessment of multi-centennial global sea level rise. In the following section, we motivate and explain the key concepts underlying the MAGICC sea level model. Section 2 covers the detailed model description and Section 3 provides key results. In Section 4, we discuss the capabilities of the presented sea level emulator and shine a first light on potential applications.

## 1.1 Motivation

Future sea level is modeled with varying degrees of complexity. Process-based modeling represents the physically most comprehensive but also computationally most expensive approach to project SLR. It is based on Atmosphere-Ocean General Circulation Models (AOGCMs) and specialized glacier, ice-sheet and ground water models that dynamically simulate sea level changes resulting from natural and anthropogenic forcings. The main sea level output from AOGCMs is the thermosteric ocean response, mostly diagnosed with post-simulation adjustments to compensate Boussinesq approximation effects (Griffies and Greatbatch, 2012). Process-based glacier and ice-sheet models are generally run separately or 'offline' and receive important boundary conditions either from observational data, AOGCMs or regional climate model input (Rae et al., 2012; Pattyn et al., 2012). Due to the complexity of the physical processes required to capture the dynamical response of each individual component, this SLR modeling approach is not feasible for efficient multi-centennial and multi-scenario research designs. It is mainly used to improve our physical understanding of the individual SLR components. The need for more efficient tools to project long-term SLR has led to the development of alternative approaches. In 2007, Semi-Empirical Models (SEMs) were introduced. They estimate sea level changes using a statistical relationship between the evolution of global mean temperature or radiative forcing and observed global mean sea level changes (Rahmstorf, 2007; Vermeer and Rahmstorf, 2009; Jevrejeva et al., 2010). SEMs do not calculate sea level based on physical processes but represent total sea level as a simplified climate



system response that is calibrated against observed changes. This approach generated considerable scientific debate and was not included in latest IPCC estimates (Orlic and Pasaric, 2013; Storch et al., 2008; Church et al., 2013a). The computational efficiency of this method, however, made it attractive to applied research questions, like investigating the global mean SLR response for different climate targets (Schaeffer et al., 2012). Recently, this method has been developed further and was ap-

plied to individual sea level components (Mengel et al., 2016). To provide an alternative to SEMs, sea level emulators have been developed to synthesize process-based sea level dynamics by calibrating simplified parameterizations to complex model projections for the main sea level contributions (Perrette et al., 2013; Schleussner et al., 2015). Progress in the understanding of individual sea level processes and the availabilty of revised sea-level estimates require sea level emulators to be updated regularly. We are able to complement the existing emulators with an up-to-date platform that consists of a more comprehen-

sive set of individual sea level components. The MAGICC sea level model represents the first efficient sea level emulator that dynamically calculates thermal expansion with a hemispheric upwelling-diffusion model based on full hemisperic ocean temperature profiles calibrated with data from Phase 5 of the Coupled Model Intercomparison Project (CMIP5) (Taylor et al., 2012). It mimics process-based sea level responses for the main seven sea level components with robust parameterizations that extend global sea level projections to 2300. Integration of the sea level model into MAGICC ensures a consistent treatment

of future sea level rise and its uncertainties along the full chain from emissions to atmospheric composition, to temperature to sea level. With the option to run large ensembles in a probabilistic setup, the MAGICC sea level model allows to explore the scenario and model uncertainty space and directly investigate SLR responses associated with mitigation pathways that are not covered by the standard RCP scenarios (Moss et al., 2010). In addition, the MAGICC global SLR projections could be used for calculating regional SLR information by using them as input for pattern scaling approaches (Perrette et al., 2013).

## 2   Model description

The MAGICC sea level emulator (Figure 1) has been developed as an extension to the widely used MAGICC model version 6 (Meinshausen et al., 2011a). The MAGICC ocean model has been revised and calibrated with available CMIP5 ocean temperature and thermal expansion data. The updated MAGICC ocean provides the basis for our thermal expansion parameterization based on Lorbacher et al. (2015). Parameterizations for global glacier, Greenland Surface Mass Balance (SMB), Antarctic

SMB, and Greenland Solid Ice Discharge (SID) have been calibrated with selected process-based projections for the corresponding SLR components. The linear response function approach for the Antarctic SID component presented in Levermann et al. (2013) was adapted to satisfy MAGICC model specifications. In addition, we have implemented the option to include land water SLR contribution estimates based on Wada et al. (2012) and Wada et al. (2016), with an extension until 2300.

### 2.1   MAGICC ocean model update and thermal expansion

MAGICC is based on a hemispheric upwelling-diffusion entrainment ocean model with depth-dependent areas for each of its 50 ocean layers (Meinshausen et al., 2011a). In this study, we provide a first series of updates for MAGICC version 7 which will be consistent with the ensemble output of CMIP5 (Taylor et al., 2012). The upwelling velocity is variable in

MAGICC and the model conserves the upwelling mass flux through layer specific entrainment which is proportional to the area decrease from top to bottom of each layer. To avoid the overestimation of ocean heat uptake for higher warming scenarios, the ocean routine includes a warming-dependent vertical diffusivity term which leads to reduced heat uptake efficiency for higher warming (Meinshausen et al., 2011a). In MAGICC6, the air temperature increases were assumed proportional to the mixed-layer ocean temperatures. The proportionality constant $\alpha$ (default value: 1.25) accounted for diminishing sea ice extent in the Arctic, exposing a larger area of the (relatively warm) surface ocean waters as warming progresses with time. Here, we replace this constant factor by a term which takes into account that this amplifying effect will itself diminish as the Arctic sea ice retreat is bound by the limit of a sea-ice free ocean in summer. The chosen functional form initially assumes a simple linear amplification (as in MAGICC6), and then progresses asymptotically towards a constant offset between the surface air temperature and top ocean layer warming. This new exponential adjustment term relates hemispheric air temperature change $\Delta T_{xA}$ to hemispheric mixed-layer ocean temperature change $\Delta T_{xO,1}$ as follows:

$$\Delta T_{xA} = \Delta T_{xO,1} + \eta(1 - e^{-\gamma \Delta T_{xO,1}}) \tag{1}$$

For large $\gamma \Delta T_{xO,1}$, the new sea-ice adjustment term moves towards a constant offset $\eta$ between surface air temperature warming $\Delta T_{xA}$ and mixed-layer ocean warming $\Delta T_{xO,1}$. However, the surface air temperature warming initially approximates $\Delta T_{xA} = \Delta T_{xO,1}(1 + \eta\gamma)$ for small $\gamma \Delta T_{x0,1}$, with $(1 + \eta\gamma)$ representing the old MAGICC6 proportionality coefficient $\alpha$. The sea-ice adjustment parameters $\eta$ and $\gamma$ are optimized together with other selected parameters for every CMIP5 model included in the MAGICC ocean model calibration (see Section 2.6). The parameter sets are optimized to represent the depth-dependent potential ocean temperature (*thetao*) responses from 37 CMIP5 models (see Table 1). The tuned model captures key features of the individual ocean heat uptake and vertical redistribution behavior of every CMIP5 model used in the calibration ensemble. Net ocean heat uptake can be robustly translated into thermal expansion (Kuhlbrodt and Gregory, 2012). Therefore, we can define the thermosteric response as the vertical sum of the layer-specific *thetao* anomalies multiplied by a corresponding thermal expansion coefficient $\alpha$ which is weighted by the specific ocean layer area. The thermal expansion coefficient $\alpha$ captures all relevant properties of seawater (potential seawater temperature, salinity, and pressure) that determine the corresponding sea level response (Griffies et al., 2014). For MAGICC, a simplified thermal expansion coefficient representation was developed which is solely based on *thetao* and pressure (Raper et al., 2001; Wigley et al., 2009). Recently, Lorbacher et al. (2015) have updated this parameterization to match CMIP5 thermal expansion behavior. We build our parameterization on Lorbacher et al. (2015) and calculate the thermal expansion coefficients for every MAGICC depth with the following polynomial of $\theta$ and $p$:

$$\alpha = (c_0 + c_1\theta_0(12.9635 - 1.0833p) - c_2\theta_1(0.1713 - 0.019263p) + c_3\theta_2(10.41 - 1.338p) + c_4p - c_5p^2)x10^{-6} \tag{2}$$

The hemispheric layer specific *thetao* values $\theta_z$ are processed for every time step with $\theta_0 = \theta_z$, $\theta_1 = \theta_0^2$, and $\theta_2 = \frac{\theta_0^3}{6000}$, assuming a mean maximum ocean depth of 6000 m. The ocean depth profile, z, is translated into the pressure profile $p = 0.0098(0.1005z + 10.5exp(\frac{-1.0z}{3500} - 1.0)$, with 3500 m as the mean ocean depth. For each of the 37 MAGICC CMIP5 ocean parameter sets, the corresponding calibration parameters $c_{0-5}$ are taken from Table S2 in Lorbacher et al. (2015). It is the combination of the CMIP5 MAGICC ocean update with the matching thermal expansion parameters that allows us to estimate 37





unique thermal expansion responses based on the selected ensemble of CMIP5 models. Our method does not cover all spatial heterogeneity effects of thermal expansion that is computed based on the three-dimensional CMIP5 fields. Therefore, we apply a model-specific scaling coefficient $\phi$ to the thermosteric estimates for each ocean layer to further improve the fit between the aggregated thermal expansion from the calibrated MAGICC ocean model and the CMIP5 thermosteric SLR (*zostoga*) estimates

(see Section 2.6 for more details).

## 2.2 Global glaciers

Mountain glaciers superseded thermal expansion as the biggest single contribution to SLR by the middle of the 20th century (Gregory et al., 2013a). The global mass balance of glaciers likely turned negative in the 19th century already, e.g. Leclercq et al. (2011). 20th century glacier mass loss contributed around 0.1 m of global sea level (Marzeion et al., 2012), with an increasing

fraction of the glacier mass loss related to anthropogenic climatic warming, reaching around 70% in recent years (Marzeion et al., 2014). Analyses of the remaining glacier mass susceptible to melt vary from around 0.35 m Sea Level Equivalent (SLE) (Grinsted, 2013) to almost 0.5 m SLE (Marzeion et al., 2012), with both studies including peripheral glaciers of the ice sheets. The latter study is based on a glacier surface mass balance model forced with regional monthly precipitation and temperature data. Changes in glacier volume are derived with the help of volume-area scaling methods. In the follow-up study (Marzeion

et al., 2014), 2300 estimates of transient glacier mass dynamics forced by 15 CMIP5 temperature and precipitation fields were complemented by equilibrium global glacier projections in response to long-term warming levels from 1 °C to 10 °C. These two experimental setups projecting transient and equilibrium glacier SLR contributions form the basis of the glacier component that has been implemented in the MAGICC sea level model. We include the Randolph Glacier Inventory 4.0 (RGI 4.0) updates on regional glacier mass loss (Pfeffer et al., 2014). The selected parameterization is based on the assumption that global glacier

melt is proportional to the remaining volume susceptible to melt (at the current global temperature) times the melt forcing. This melt forcing is expressed by the temperature difference between current temperature and the temperature that would be expected if the currently remaining glacier volume was in equilibrium. Thus, we apply the following functional form to relate the global glacier SLR response $GL_t$ to the remaining global glacier volume as well as the temperature forcing:

$$GL_t = GL_{t-1} + \kappa(V_{eq}(T_t) - V_{cum})(T_t - T_{eq}(V_{cum}))^\nu \qquad (3)$$

, with calibration parameters $\kappa$ and $\nu$ and $V_{eq}(T_t)$ being the equilibrium glacier volume change that would result from warming level $T_t$. This value is interpolated from the Marzeion et al. (2014) glacier equilibrium response data. $V_{cum}$ is the cumulated glacier volume change since the year 1850. $T_{eq}(V_{cum})$ is the inverse function of the equilibrium glacier response $V_{eq}$ to $T_t$ and gives the temperature that would lead to the glacier volume change $V_{cum}$ in terms of a theoretic equilibrium response.

## 2.3 Greenland ice sheet

The Greenland contribution to SLR increased rapidly during the last decades of the 20th century (Vaughan et al., 2013). Regional atmospheric and ocean warming has triggered wide spread surface melt (Fettweis et al., 2011) and solid ice discharge (Joughin et al., 2012). An increasingly negative SMB and a growing SLR contribution from SID, which captures accelerating



ice stream flow and more frequent calving events due to warmer ocean temperatures, have been identified to be responsible for about half of the observed mass loss each (van den Broeke et al., 2009; Khan et al., 2015). The Greenland ice sheet is expected to become one of the largest SLR contributions in the future (Huybrechts et al., 2011), with potentially irreversible ice sheet loss for scenarios of persistent and strong warming (Robinson et al., 2012; Levermann et al., 2013). In the following,

we present SMB and SID parameterizations that have been implemented and calibrated in the MAGICC sea level model.

### 2.3.1 Surface mass balance

The mass balance at the surface of the Greenland ice sheet is predominantly determined by the accumulation of snowfall in winter and runoff through melting in summer. Continuing global warming will influence the SMB through both increased snowfall and increased melting (Gregory and Huybrechts, 2006). As melting is expected to increase stronger than snowfall,

SMB losses will likely dominate future Greenland contributions to SLR (Church et al., 2013a; Goelzer et al., 2013). Regional surface air temperatures are the primary driver of these projected SMB changes if we assume future precipitation changes over Greenland to be scalable with rising temperatures (Fettweis et al., 2013; Frieler et al., 2012). Regional atmospheric temperatures are closely linked to the global mean surface air temperature $tas$. We utilize this link for our sea level component by relating two $tas$ dependent terms to capture the long-term SMB sea level response. In the parameterization, the SMB response to $tas$

can vary from either being approximated as scaling linearly, or non-linearly with exponent $\varphi$, or as a combination of both. The calibration procedure chooses the optimal balance of the linear and non-linear terms. Furthermore, the surface melt contribution is dampened by diminishing ice availability for high warming scenarios and eventually becomes zero when all available ice is melted. Hence, the cumulated Greenland SMB SLR contribution $GIS_t^{SMB}$ at time step $t$ can be written as:

$$GIS_t^{SMB} = GIS_{t-1}^{SMB} + \upsilon(\chi T_t + (1-\chi)T_t^{\varphi})\left(1 - \frac{GIS_{t-1}^{SMB}}{GIS_{max}^{SMB}}\right)^{0.5} \tag{4}$$

The maximum Greenland ice volume available for surface melt $GIS_{max}^{SMB}$ is about 7.36 m (Bamber et al., 2013). The overall temperature sensitivity is denoted by $\upsilon$ and the choice of $\varphi$ sets the degree of non-linearity, while $\chi$ determines the relative magnitude of the linear and nonlinear terms. We calibrate the three parameters $\upsilon$, $\chi$, and $\varphi$ with reference data from Fettweis et al. (2013). Their process-based Greenland SMB projections until 2100 are based on the regional climate model Modele Atmospherique Regional (MAR) which is coupled to the Soil Ice Snow Vegetation Atmosphere Transfer (SISVAT) scheme.

The MAR model is forced by CMIP5 data for temperature, wind, humidity, and surface pressure. Comparing the MAGICC Greenland SMB response to millenial projections of Greenland ice sheet sea level contributions (Huybrechts et al., 2011; Goelzer et al., 2012) indicates that the functional form of our SMB parameterization will hold for multi-centennial projections at least until 2300.

### 2.3.2 Solid ice discharge

Future ocean warming is expected to reduce the frontal stress of the Greenland outlet glaciers while increased melt water from atmospheric warming can reduce the friction at the bottom of these glaciers. Both processes lead to the speed up and thinning



of these glaciers, with increased discharge of solid ice into the oceans (Nick et al., 2009). Even though the SMB contribution is projected to dominate the Greenland contribution to SLR, the SID component has the potential to contribute significantly to SLR (Jacobs et al., 1992; Rignot et al., 2010; Joughin et al., 2012). Recent attempts to quantify the future ice-dynamic SLR contribution for Greenland vary widely, mainly due to different methodologies (Nick et al., 2013; Vizcaino et al., 2015; Fürst

et al., 2015). We select one of the key approaches presented in the latest IPCC assessment for our reference data (Church et al., 2013a): Nick et al. (2013) use flowline modeling to project mass loss from Greenland's four main outlet glaciers Helheim, Jakobshavn Isbrae, Kangerdluqssuaq, and Petermann until 2200. The model is forced with ocean and atmosphere data from SRES A1B and RCP8.5 scenario runs conducted with the CMIP3 model ECHAM5-OM. As the four main outlet glaciers drain about 20% of the entire Greenland ice sheet area, the sum of the individual glacier contributions has been multiplied by a factor

of 5 to estimate the SID sea level contribution of the whole ice sheet (Church et al., 2013a; Price et al., 2011). We use the same approach to emulate the response of Nick et al. (2013), with the cumulated Greenland SID SLR contribution $GIS_t^{SID}$ at time step $t$ being:

$$GIS_t^{SID} = s(GIS_{max}^{outlet} - GIS_{Vdis(t)}^{outlet}) \qquad (5)$$

, with $GIS_t^{SID}$ defined as the difference of the initial maximum ice volume susceptible to discharge and the remaining ice

volume available for dischareg at time step $t$. Maximum ice volume, $GIS_{max}^{outlet}$, and remaining ice volume at time step $t$, $GIS_{Vdis(t)}^{outlet}$, are determined for the four main Greenland outlet glaciers. By applying the scaling factor $s = 5$, the sea level contribution is then scaled up to the entire Greenland ice sheet. For $t = 0$, $GIS_{Vdis(t=0)}^{outlet} = GIS_{max}^{outlet}$. The remaining ice volume susceptible to discharge at time step $t$, $GIS_{Vdis(t)}^{outlet}$, has the following function form:

$$GIS_{Vdis(t)}^{outlet} = GIS_{Vdis(t-1)}^{outlet} - max(0, \varrho GIS_{Vdis(t-1)}^{outlet} e^{\epsilon T(t-1)} + \zeta) \qquad (6)$$

, with the annual discharge being the product of the discharge sensitivity $\varrho$, the SID volume $GIS_{Vdis(t-1)}^{outlet}$ available at time step $t-1$, and an exponential *tas* term which is dependent on a temperature sensitivity $\epsilon$, plus the constant $\zeta$. We have calibrated $\varrho$, $\epsilon$, $\zeta$, and the maximum SID outlet glacier volume $GIS_{max}^{outlet}$ based on the projected minimum and maximum contributions for dynamic retreat and thinning for scenarios SRES A1B and RCP8.5, shown in Figure 3e of Nick et al. (2013). An upper limit of the potential Greenland SID discharge contribution has not been clearly defined yet (Goelzer et al., 2013; Price et al.,

2011). We include the maximum SID outlet glacier volume susceptible to discharge $GIS_{max}^{outlet}$ in our calibration. Applying the scaling suggested by Church et al. (2013a), our total Greenland SID maximum ice discharge volumes amount to around 202 mm and 268 mm SLE for the minimum and maximum cases presented in Nick et al. (2013). For comparison, Winkelmann and Levermann (2012) obtained 420 mm for the ice-dynamic Greenland sea level contribution, indicating, however, that the actual amount might be significantly smaller. For high warming scenarios, our SID projections deplete $GIS_{max}^{outlet}$ before the

30  year 2300 which causes the annual Greenland SID sea level contribution to drop to zero.

## 2.4 Antarctic ice sheet

Air temperatures over the Antarctic ice sheet are generally much colder than over the Greenland ice sheet. They will be too low to cause wide-spread surface melting, even under strong global warming (Church et al., 2013a). Only peripheral, low-lying





glaciers, especially around the Antarctic Peninsula are susceptible to retreat through increased ablation (Krinner et al., 2006). A warmer atmosphere over Antarctica will however hold more moisture, leading to higher snowfall. This effect is expected to lead to a positive SMB through snow accumulation and, thus, a slightly negative SLR contribution (Bengtsson et al., 2011; Gregory and Huybrechts, 2006). The main driver of Antarctic ice loss and a resulting positive sea level contribution is the increased melting of ice shelves through warmer ocean waters (Joughin et al., 2012; Bindschadler et al., 2013). SID will be the dominant SLR contribution of Antarctica, with increasing ocean temperatures causing basal melt in marine-based ice sheet sectors, potentially even triggering marine ice-sheet instabilities and irreversible ice loss (Huybrechts et al., 2011; Joughin et al., 2014). We implemented parameterizations capturing both, the Antarctic SMB and the SID contributions to SLR in the MAGICC SLR mode. They are presented below.

### 2.4.1 Surface mass balance

Positive Antarctic SMB anomalies under all warming scenarios lead to consistently negative contributions to global sea level for the 21st century. Similar to Greenland, a strong (but different) link exists between future Antarctic SMB and global mean surface air temperature *tas*. Several studies confirmed the Clausius-Clapeyron equation based exponential relationship between atmospheric warming and SMB accumulation. The values range from 3.7 % $°C^{-1}$ (Krinner et al., 2006) up to around 7 % $°C^{-1}$ (Bengtsson et al., 2011), with most recent estimates based on a large ensemble of climate models pointing to about 5 % $°C^{-1}$ (Frieler et al., 2015). Ligtenberg et al. (2013) has been one of the few studies using regional climate simulations to assess Antarctic SMB changes beyond 2100, however without accounting for climate-ice sheet feedbacks. Their assessment is based on the regional atmospheric climate model RACMO2 (Lenaerts et al., 2012) and the two global climate models ECHAM5 (Roeckner et al., 2003) and HadCM3 (Johns et al., 2003) that have been forced by two comparably moderate emission scenarios (SRES A1B and ENSEMBLES E1), leading to a 2200 Antarctic warming of 2.4-5.3 °C. Results show SMB increases of 8-25 % which translate into a global sea level drop of 73-163 mm. We select these projections as reference for our SMB parameterization. Due to the expected strong SMB link to *tas*, we have chosen a simple functional form that relates the annual Antarctic SMB sea level contribution to this primary driver:

$$AIS_t^{SMB} = AIS_{t-1}^{SMB} + \xi(\rho T_t + (1-\rho)T_t^{\sigma}) \tag{7}$$

The annual change in the Antarctic SMB contribution to SLR is derived from the sum of a linear and non-linear *tas* term, calibrated with the three parameters $\xi$, $\rho$, and $\sigma$. The transfer from global mean *tas* to regional surface air temperature changes as well as the translation of air temperatures into snowfall accumulation is captured in $\xi$, while $\rho$ controls the non-linearity of the parameterization. The calibrated parameterization is then used to extend Antarctic SMB SLR estimates until 2300 presuming that the rationale behind the projections presented in Ligtenberg et al. (2013) hold for another 100 years. This is consistent with findings from up to 3000 year long Antarctic SMB simulations that are forced by idealized scenarios doubling or quadrupling atmospheric $CO_2$ concentration levels (Vizcaíno et al., 2010; Huybrechts et al., 2011). Results from these studies show ice mass gains due to additional snowfall for more than 500 years after the start of the experiments, e.g. see Fig. 7 in Huybrechts et al. (2011).



### 2.4.2 Solid ice discharge

Improved process understanding has allowed for a first assessment of the Antarctic dynamic ice-discharge contribution to SLR in the IPCC Fifth Assessment Report (Church et al., 2013a). Antarctic SID has the potential to supersede all other sea level contributions because of the vast ice masses accessible for warm ocean waters and susceptible to self-amplified retreat

(DeConto and Pollard, 2016). Loss of these ice masses alone would eventually lead to several meters of global SLR (Bamber et al., 2009). Recent observations and modeling suggests that the process of self-sustained retreat has already begun and will dominate over the slower adjustments to *tas* and precipitation changes across the Antarctic continent on decadal to centennial timescales (Joughin et al., 2014; Rignot et al., 2014; Favier et al., 2014). Levermann et al. (2014) convolve the responses from five different Antarctic ice-sheet models to basal melt forcing as used in the SeaRISE project (Bindschadler et al., 2013) with

a large set of MAGICC temperature projections for the full suite of RCP scenarios. In their study, the projected global mean *tas* signal is converted into subsurface ocean temperatures that is translated into basal melt forcing. The melt forcing is then convolved with individual response functions for the Amundsen Sea, Ross Sea, Weddell Sea, and East Antarctic sectors. This approach is well suited for the MAGICC sea level model implementation because it relates the ice-sheet response directly to *tas*. We implement a step-wise convolution routine in the MAGICC SLR model which allows us to process the response

functions for the different sectors. The total SLR contribution from Antarctic SID, $AIS^{SID}$, can be written as the sum of the contributions from the individual sectors:

$$AIS^{SID} = \sum_{n=1}^{4} \int_{0}^{t} F_n(\tau)R_n(t-\tau)\,\mathrm{d}\tau \tag{8}$$

The sector-specific basal melt forcing $F_n$ is the product of the basal melt sensitivity $\psi$ and the sector-specific subsurface ocean temperature anomaly $dT_{OCN}$. The region-specific ice sheet response function $R_n(t-\tau)$ is based on linear response theory

(Winkelmann and Levermann, 2012). The basal melt forcing $F$ is the product of the basal melt sensitivity $\psi$ and the sector-specific subsurface ocean temperature anomaly $dT_{OCN}$. Starting in 1850, Levermann et al. (2014) derived the latter from the projected annual MAGICC global mean *tas* anomalies via ocean temperature scaling and a time delay between surface and ocean subsurface warming. We adopt all relevant melt forcing parameters from Levermann et al. (2014). They determined these parameters either through calibrations against 19 CMIP5 models or adopted them from the existing literature, like the basal

melt sensitivities ranging from 7 $ma^{-1}K^{-1}$ to 16 $ma^{-1}K^{-1}$ (Holland et al., 2008; Payne et al., 2007; Jenkins, 1991). The response functions are derived for 500 years and cover the time frame of their source experiments described in Bindschadler et al. (2013). We provide Antarctic SID projections up to the year 2300. For the MAGICC component, only response functions from the three ice-sheet models that have an explicit representation of ice-shelf dynamics are included, namely PennState-3D (Pollard and DeConto, 2012), PISM (Winkelmann et al., 2011; Martin et al., 2011), and SICOPOLIS (Sato and Greve, 2012).

The response functions presented by Levermann et al. (2014) and implemented here do not account for all ice sheet processes and feedbacks. Thus, the Antarctic SID estimates provided by the MAGICC sea level model may underestimate the actual Antarctic SID sea level response.





## 2.5 Land water storage

The assessment of the observed and projected anthropogenic land water contribution to SLR is subject to ongoing discussions (Konikow, 2011; Pokhrel et al., 2012; Wada et al., 2012; Church et al., 2013a; Wada et al., 2016). Associated uncertainties are high, mainly due to sparse data coverage and unknown process understanding. Two major processes drive changes in land

water storage: The depletion of groundwater resources which positively contributes to SLR and water impoundment which dampens the SLR signal. Analyses show that the latter contribution has been shrinking since the late 20th century (Gregory et al., 2013b) which leaves groundwater depletion as the main human-driven Land Water Storage (LWS) SLR contribution throughout the 21st century and beyond. We include the option to provide LWS sea level estimates based on the approach introduced by Wada et al. (2012). They forced the hydrological model PCR-GLOBWB (van Beek et al., 2011) with climate

projections from AOGCMs to derive estimates for future groundwater depletion until 2100. Original estimates had to be revised because only roughly 80% of annually depleted groundwater ends up in the oceans (Wada et al., 2016). We adapt our time series accordingly, reducing the Wada et al. (2012) sea level contribution estimates from groundwater depletion by 20%. We use the 30-year average annual depletion rate for the period 2071-2100 to extend the projections beyond the 21st century. We assume that projected rates of human water use and groundwater abstraction, which show more constant rates towards the end of

the 21st century (Wada, 2015), will persist beyond 2100. The fraction of non-renewable groundwater to total groundwater abstraction is projected to increase to around 50% by 2100 (Wada, 2015). This indicates that, ultimately, the total amount of groundwater available for abstraction is limited. To account for such an upper bound of the LWS sea level contribution, we use a term that relates the cumulated LWS contribution to a theoretic maximum LWS volume that can be depleted. No distinction is made between different climate scenarios for the post-2100 LWS extension due to the limited process understanding and the

associated large uncertainties (Church et al., 2013a). Hence, we implement the revised Wada et al. (2012) estimates until 2100 and apply the following post-2100 LWS parameterization:

$$LWS_t = LWS_{t-1} + LWS_{const} \left(1 - \frac{LWS_{t-1} - LWS_{2100}}{LWS_{max} - LWS_{2100}}\right)^{0.5} \tag{9}$$

The maximum LWS volume $LWS_{max}$ has not been quantified yet de Graaf et al. (2014). However, Gleeson et al. (2015) quantified the amount of modern groundwater which is defined as less than 50 year old groundwater located in the top 2 km

of the continental crust. This type of groundwater dominates the interaction with general hydrological cycle and the climate system. It is also the most accessible for land use (Gleeson et al., 2015). We here define $LWS_{max}$ as the total amount of available modern groundwater which has been estimated to be around 350,000 $km^3$, roughly translating to 1000 mm SLE.

## 2.6 Model calibration

For the MAGICC ocean model calibration, we use two CMIP5 variables for our reference data set: ocean potential temperatures

(*thetao*) and thermal expansion (*zostoga*). Ocean depths specific *thetao* time series are extracted for a total of 37 CMIP5 models which have been running pre-industrial control (*pictrl*), historical, some or all of the RCP experiments as well as the idealized 1% CO2 per year increase (*1pctCO2*) experiments. Each individual model output is converted into hemispheric annual mean



*thetao* depth profile time series that are then vertically interpolated to match the MAGICC ocean layer depths. We combine historical and RCP runs to create layer-specific time series from 1850 to 2100 or 2300 depending on the experiment lengths of the individual CMIP5 model. Ocean temperature data available from the CMIP archives is subject to drift because the time scales for the ocean to adjust to external forcing are much longer than the length of the control experiments (Taylor et al., 2012;

Gupta et al., 2013). Individual model drifts have been identified based on the respective *pictrl* runs. The full linear trend from the *pictrl* experiments has been removed from the historical plus RCP and *1pctCO2* scenario time series.

The initial *thetao* profiles are prescribed for every CMIP5 model calibration as well as the respective depth-dependent ocean area fractions. We incorporate *zostoga* estimates for each of the 37 CMIP5 ensemble members by detrending the times series with the full linear trend of the *pictrl* runs. To ensure a full CMIP5-consistent calibration setup, we constrain MAGICC for

every CMIP5 model optimization by prescribing the corresponding model-specific annual global mean surface air temperature *tas*. We select a total of 9 parameters for the calibration. The vertical thermal diffusivity, $K_z$, its sensitivity to global-mean surface temperatures at the mixed layer boundary, $\frac{dK_{z_{top}}}{dT}$, the sea-ice adjustment parameters $\eta$ and $\gamma$ described above, the initial upwelling rate $w_0$, the ratio of changes in the temperature of the entraining waters to those of the polar sinking waters $\beta$, the ratio of variable to fixed upwelling for every time step $\frac{\Delta w_t}{w_t}$, and the corresponding threshold temperatures which lead to

constant upwelling rates, namely $T_{w_t}$, and the global thermal expansion scaling coefficient $\phi$. More details on the individual parameters can be found in Meinshausen et al. (2011a) except for the sea-ice adjustment variables described in Section 2.1. For every CMIP5 model, this suite of calibration parameters is optimized based on the scenario specific CMIP5 *thetao* data for the representative layers 1 (30m layer mean depth), 2 (110m), 3(210m), 8 (710m), 15 (1410m), 30 (2910m), and 40 (3910m), and the corresponding *zostoga* time series. The eight calibration layers have been selected to allow the MAGICC ocean model to

emulate the key features of the CMIP5 ocean temperature profiles, with the majority of calibration layers set in the upper ocean to ensure sufficient coverage of the stronger temperature gradients. The number of reference layers is not increased further to preserve computational efficiency. 5000 random parameter sets are drawn for every model optimization. The resulting best fit is subsequently used for the initialization of the automated Nelder-Mead simplex optimization routine (Nelder and Mead, 1965) with a termination tolerance of $10^{-8}$ and a maximum iteration number of 10,000. We use weighted Residual Sum of

Squares (RSS) for Goodness-Of-Fit (GOF) diagnostics during the optimization process (Meinshausen et al., 2011a). The ocean calibration also takes into account the available CMIP5 *zostoga* time series. The *zostoga* optimization component only receives three orders of magnitude less relative weight than the *thetao* component in order to prioritize the accurate layer-by-layer emulation of the respective CMIP5 model *thetao* time series. The GOF values are then divided by the number of calibrated model years, accounting for the varying amount of scenario data available for each model. This allows us to compare the GOFs

of the calibrations for all 37 CMIP5 models.

The calibration procedures for the other SLR components also optimize the specific parameters listed in Tables 2 to 5 based on the Nelder-Mead Simplex method with a termination tolerance of $10^{-8}$ for a change in RSS during the last iteration. All the remaining SLR components use reference SLE contributions in millimeters for the respective optimizations. For the glacier contribution, the MAGICC sea level response is fitted to the transient Marzeion et al. (2014) projections. The free parameters $\kappa$

and $\nu$ are calibrated for each of the 14 CMIP5 reference models and their respective combined historical and RCP simulations,





starting in 1850. Corresponding CMIP5 global mean *tas* projections are prescribed in the MAGICC model to ensure consistency with CMIP5. We use a subset of the model specific 1965-2100 projections made available by Fettweis et al. (2013) to calibrate the parameterization for the Greenland SMB contribution. 24 CMIP5 models are selected based on the availability of CMIP5 *tas* projections for the scenarios RCP4.5 and RCP8.5. We then prescribe these global mean *tas* time series for the calibration

procedure of the three parameters $\upsilon$, $\chi$, and $\varphi$. Calibration data for the Greenland SID component is only available for one GCM, ECHAM5. For the optimization of the parameters $\varrho$, $\epsilon$, $\zeta$, global mean *tas* runs for SRES A1B and RCP8.5 are used with 2200 extensions, repeating the last decade of the 21st century ten times (Nick et al., 2013). The calibration of the Antarctic SMB component is based on process-based SLR responses forced by two GCMs (Ligtenberg et al., 2013). In this reference study, the ECHAM5 and HadCM3 model output was applied for scenarios SRES A1B and ENSEMBLES E1. We replicate

these GCM responses and use the provided Antarctic SMB sea level contributions starting in 1980 to determine the optimal parameters $\xi$, $\rho$, $\sigma$. The Antarctic SID as well as the LWS components are not subject to calibration procedures as they apply the same method of the reference study in case of Antarctic SID or simply include and extend the reference data for LWS.

## 3 Results

The MAGICC ocean model update yields optimal parameter sets for every CMIP5 model used in the calibration procedure

outlined above. Those sets are listed in Table 1. In Figure 2, we show both the 90% model range and median for the reference CMIP5 global potential ocean temperature anomalies as well as the median MAGICC global ocean warming profile averaged over 2081 to 2100 relative to the reference period 1986 to 2005. MAGICC is able to capture the key CMIP5 features for all RCP scenarios. The median model response either matches or is close to the median of the CMIP5 responses. However, the updated MAGICC ocean shows two distinct deviations from the CMIP5 data. First, the mid-ocean layers between the layers

15 (1410 m) and 30 (2910 m) are colder than the CMIP5 reference data. Second, the bottom ocean layers tend to be warmer as for the corresponding CMIP5 models in all RCP scenarios. Both of these features can be explained by the upwelling-diffusion design of the MAGICC model. Mid-layer warming plumes of specific CMIP5 models like the GISS-E2-R mode cannot be captured by the parameterization while the warmer MAGICC bottom water is maintaining the balance between upwelling and the heat entrainment through downwelling (see also Meinshausen et al. (2011a), section A4). Besides these caveats, the

MAGICC ocean component captures the hemispherically averaged CMIP5 ocean warming for the different RCP scenarios well. We derive CMIP5 consistent thermal expansion estimates based on the optimal ocean parameter sets and the additional thermal expansion scaling parameter $\phi$ (see Table 1).

In Figure 3, we synthesize the calibration results for all sea level contributions captured by the MAGICC sea level model. Panels (a) to (d) show the model specific global thermal expansion responses and the corresponding CMIP5 *zostoga* reference

data for the four RCP scenarios. The number of available reference runs differs for each scenario as does the length of the simulations. The updated MAGICC ocean component is able to mimic the CMIP5 thermal expansion time series. Relative to 1850, the calibration yields a 2100 thermosteric SLR range of 101 to 244 mm (CMIP5: 113 to 231 mm) for RCP2.6, 149 to 302 mm (161 to 290 mm) for RCP4.5 , 163 to 321 mm (174 to 309 mm) for RCP6.0, and 241 to 474 mm (261 to 445 mm)





for RCP8.5. The corresponding 2300 thermosteric SLR responses range from 192 to 335 mm for RCP2.6 (CMIP5: 180 to 288 mm), 327 to 678 mm for RCP4.5 (345 to 707 mm), 614 to 715 mm for RCP6.0 (635 to 658 mm), and 1018 to 1793 mm for RCP8.5 (1040 to 1909 mm). In contrast to some detrended *zostoga* CMIP5 model time series, the MAGICC thermal expansion projections do not show negative slopes in the 20 century which is consistent with observations ((Church et al., 2013b)).

5  The calibrated global glacier SLR response and the corresponding reference data are shown in panels (e) to (h), the specific calibration result are listed in Table 2. The MAGICC projections show good agreement with the updated Marzeion et al. (2012) data (Fig. 3, panels (e) to (h)). Relative to 1850, the estimated glacier SLE contributions in 2100 are 146 to 258 mm (Marzeion et al.: 133 to 257 mm) for RCP2.6, 162 to 271 mm (159 to 277 mm) for RCP4.5, 163 to 272 (163 to 276 mm) for RCP6.0, and 181 to 295 mm (198 to 308 mm) for RCP8.5. For 2300, projected SLR from glaciers amounts to a SLE range of 175 to 305
10  mm (Marzeion et al.: 189 to 305 mm) for RCP2.6, 258 to 372 mm (254 to 359 mm) for RCP4.5, and 323 to 437 mm (338 to 444 mm) for RCP8.5.

In panels (j) and (k), we cover the Greenland SMB contribution, both the reference data from Fettweis et al. (2013) and the sea level model estimates based on the optimal paramester sets shown in Table 3. Our model shows high agreement with the reference data. For 2100, we project SLE ranges from 15 to 117 mm (Fettweis et al.: 17 to 114 mm) based on RCP4.5 and
SLE ranges from 40 to 209 (48 to 206 mm) based on RCP8.5. Projections start in 1965, being the first year of the calibration data. The Greenland SID calibration results are depicted in panels (l) and (m) of Figure 3. We show MAGICC sea level model estimates based on the calibration results listed in Table 4. As presented by Nick et al. (2013), we show projections of the minimum and maximum cases for the combined contribution from the four major outlet glaciers prior to up-scaling to the entire Greenland ice sheet. Estimates are provided relative to the year 2000. For the SRES A1B scenario, the SLE projections
range from 15 to 25 mm (Nick et al.: 14 to 24 mm) for the last year of the available reference data in 2190. For the same year, we project 26 to 43 mm (26 to 43 mm) based on the RCP8.5 scenario.

Calibration results for the Antarctic SMB component which negatively contributes to future SLR are listed in Table 5. Corresponding output is shown in panels (n) and (o). Starting in 1980, the reference data from Ligtenberg et al. (2013) provides projections that go beyond 2100 only for the model HadCM3. For the ENSEMBLES E1 scenario, the two model specific 2100
SLE responses range from -29 to -18 mm (Ligtenberg et al.: -27 to -20 mm). The 2200 estimate lies at -66 mm (-73 mm) based on the HadCM3 parameter set. The 2100 values for the SRES A1B scenario span from -50 to -33 mm (Ligtenberg et al.: -43 to -32 mm), while the 2200 Antarctic SMB SLE response is projected to be -158 mm (-163 mm). As we model the Antarctic SID sea level component with the linear response function approach presented by Levermann et al. (2014), it is not calibrated against any reference data. The MAGICC component utilizes the responses from the three ice-sheet models of that study which
include an explicit representation of ice-shelf dynamics. As the sea level responses for this subset of ice-shelf models are not available, we show the 90% model range and the median of all five ice-sheet models from Levermann et al. (2014) in panels (p) to (s) of Figure 3. CMIP5 model specific parameter sets have been determined for the three different ice-shelf models (Levermann et al. (2014), Tables 2-5). In 2100, the 90% ranges of the MAGICC responses based on the ice-shelf model subset correspond to 44 to 282 mm SLE (Levermann et al.: 15 to 227 mm) for RCP2.6, 53 to 282 mm (17 to 267 mm) for RCP4.5,
59 to 272 mm (17 to 277 mm) for RCP6.0, and 70 to 358 mm (20 to 365 mm) for RCP8.5. For 2300, 90% of the MAGICC





projections lie within 269 and 664 mm SLE (Levermann et al.: 69 to 635 mm) for RCP2.6, 312 and 1123 mm (119 to 1182 mm) for RCP4.5, 590 and 1197 mm (161 to 1719 mm) for RCP6.0, and 953 and 2450 mm (300 to 3535 mm) for RCP8.5, respectively. The MAGICC Antarctic SID estimates, which are based on the physically more complex ice-shelf models only, mostly lie within the 90% range of Antarctic SID sea level contributions provided by Levermann et al. (2014).

In panel (t), we show SLE responses for the scenario independent land water SLE component. Until 2100, we include the net land water SLE contribution as presented in Figure 3 of Wada et al. (2012), corrected by the 20% fraction of land water that does not reach the global ocean Wada et al. (2016). Post 2100, we assume a constant annual contribution based on the assumptions outlined in section (2.5). 2100 estimates span a global sea level contribution of 39 to 77 mm. The extended land water projections range from 156 to 261 mm SLE for 2300.

With the individual SLR components calibrated, we can project total SLR as the combination of the individual SLE responses from each of the seven sea level components. Two different MAGICC setups are used to project global SLR until 2100 and 2300 based on the four RCP scenarios and their extensions. The ocean model update is not sufficient to make the MAGICC model fully CMIP5 consistent because other crucial climate system components like the carbon cycle have not been updated yet. To overcome this issue, we constrain the MAGICC model with available CMIP5 global mean *tas* time series. Together

with the corresponding calibrated MAGICC ocean model parameter sets, we are able to create a CMIP5 environment that allows us to compare our 2100 global SLR projections to the latest IPCC estimates. Beyond 2100, the number of available CMIP5 simulations is much smaller, with only two 2300 model runs available for RCP6.0, for example. In order to also provide a sufficiently large number of model runs for 2300, we use a historically-constrained, probabilistic MAGICC design (Meinshausen et al., 2009). This 600 member ensemble has been previously applied to capture the climate sensitivity range of

the latest IPCC assessment (Rogelj et al., 2012, 2014). For this second setup, MAGICC is not forced to match CMIP5 global mean *tas*, allowing us to provide consistent ensemble projections out to 2300. For this ensemble, we randomly draw from the CMIP5 ocean model parameter sets and the calibration results for each sea level model component. Random samples are also sourced between the minimum and maximum realizations for the Greenland SID and LWS component as well as between the empirical basal melt sensitivities for the Antarctic SID contribution (Levermann et al., 2014). For consistency, we adopt the

same ensemble size for the CMIP5 constrained MAGICC setup and randomly select the specific CMIP5 global mean *tas* time series in addition to the other randomized parameter sets from the individual sea level components.

In Table 6, we show median SLR estimates for the 2081 to 2100 average and 66% ranges for every individual component, with corresponding IPCC reference estimates and likely ranges. The individual MAGICC sea level contributions are in good agreement with the IPCC estimates. Figure 4 shows the full suite of MAGICC SLR projections for the RCP scenarios. The

smaller panels (a) to (d) give 90% and 66% ranges as well as median responses for all RCP scenarios until 2100 based on the CMIP5 consistent setup. Additional bars are provided for the IPCC reference data and the probabilistic MAGICC setup which is not constrained to CMIP5. For the CMIP5 consistent MAGICC setup, 2100 median SLR is projected to be 0.47 m (66% range: 0.38 m to 0.59 m) for RCP2.6, 0.56 m (0.47 m to 0.68 m) for RCP4.5, 0.58 m for (0.47 m to 0.72 m) for RCP6.0, and 0.79 m (0.60 m to 0.96 m) for RCP8.5 (see also Table 7). All SLR projections are provided relative to the reference period

1986 to 2005. MAGICC SLR estimates for 2100 are generally higher than the IPCC projections. CMIP5 consistent projections





of average 2081 to 2100 SLR lie well withing the IPCC range, with median estimates on average 0.04 m higher than the corresponding IPCC values (Church et al., 2013a). In panel (e), we provide 2300 SLR projections for the RCP extensions based on the probabilistic MAGICC setup which is not constrained to CMIP5. For RCP2.6, the median SLR reponse is 0.97 m (66% range: 0.76 to 1.35 m). We project a median of 1.65 m (1.23 to 2.37 m) for RCP4.5, 2.32 m (1.67 to 3.41 m) for RCP6.0,

and up to 5.11 m (3.63 to 7.55 m) for RCP8.5 (see also Table 7). The global 2300 SLR responses are provided for all RCPs and each sea level component in the Appendix Figures A1 to A4.

Figure 5 shows the global mean *tas* responses based on the historically-constrained, probabilistic MAGICC setup, which is used for the 2300 SLR projections. Each panel also includes the available CMIP5 global mean *tas* time series. 2300 MAGICC median global mean *tas* fall well within the available CMIP5 range for RCP4.5, RCP6.0 and RCP8.5. The MAGICC median

global mean *tas* response is at the lower end of 2300 CMIP5 temperatures for RCP2.6. For this scenario, the cooling over 22nd and 23rd centuries is consistent with previous MAGICC studies, e.g. Meinshausen et al. (2011b). The overall historically-constrained, probabilistic MAGICC global mean *tas* response for 2100 is stronger than the CMIP5 reference data for RCP4.5, RCP6.0 and RCP8.5 scenarios, with MAGICC median global mean *tas* estimates being at the high end of corresponding RCP6.0 and RCP8.5 CMIP5 projections. These higher 2100 global mean *tas* signals are also reflected in the corresponding

MAGICC 2100 SLR estimates, given the strong air temperature dependence of the sea level model (see panels (a) to (d) of Figure 4).

## 4   Discussion

The MAGICC sea level model presented here synthesizes long-term sea level projections for seven sea level components and provides up-to-date and efficient representations of the individual SLR contributions, validated against process-based model

results (see Figure 3 and Section 2). Thermal expansion is calculated with an updated version of the MAGICC hemispheric upwelling-diffusion ocean model and an ocean-layer specific thermal expansion parameterization by Lorbacher et al. (2015). We are therefore able to directly account for ocean heat uptake effects, which is an advantage over other contribution-based approaches that simply derive thermal expansion from global mean air temperature changes (Mengel et al., 2016). The MAGICC ocean thermal expansion component is calibrated to be fully consistent with CMIP5. The glacier component parameterization

accounts for both transient projections of glacier mass loss (Marzeion et al., 2012) and equilibrium glacier responses based on Marzeion et al. (2014). The SMB and SID parameterizations for both ice sheets reflect available process-based reference data (Fettweis et al., 2013; Nick et al., 2013; Ligtenberg et al., 2013; Levermann et al., 2014). In addition, new process understanding has been included in the land water component (Wada et al., 2016). The full MAGICC model, including the sea level module, can be run in less than one second for 100 model years on a single core. This makes it an efficient platform to provide

large ensembles of global sea level projections.

Projecting SLR beyond 2100 and providing physically-consistent global estimates out to 2300 has been one of the key motivations for the development of the MAGICC sea level model. For five of the seven sea level components, the reference data used for calibrating the individual contributions extends beyond 2100. For thermal expansion, global glacier, and Antarctic





SID contribution, the reference calibration period spans from 1850 to 2300. The remaining components are based on physically plausible assumptions, which allow us to also provide 2300 estimates, assuming that the calibrated parameterizations for each sea level component remain valid. The fact that our sea level model is directly calibrated against long-term projections and reproduces these projections well makes us confident that the model contributes to advancing efficient long-term sea level

projections and that it is superior to, for example, the recently published 'constrained-extrapolation' approach by Mengel et al. (2016).

Both CMIP5 ocean and air temperatures serve as input for the presented sea level model. Other published sea level emulators only utilize air temperature projections, also provided by MAGICC, either based purely on available CMIP3 calibration results (Meinshausen et al., 2011a; Perrette et al., 2013) or an updated historically-constrained probabilistic MAGICC setup that

reflects the latest IPCC climate sensitivity estimates (Schleussner et al., 2015; Mengel et al., 2016). We here provide the first major step to making MAGICC fully CMIP5 consistent, with the ocean model now emulating 37 CMIP5 hemipsheric potential ocean temperature and thermal expansion responses. However, other crucial elements of the MAGICC model, like the atmosphere and the carbon cycle, are not yet calibrated to CMIP5. When combining the CMIP5-calibrated ocean with the older atmosphere and carbon cycle calibrations, the resulting air temperatures are higher than CMIP5 (see Figure 5). To ensure

a robust MAGICC sea level model, the individual components were either calibrated with prescribed CMIP5 temperatures, or with CMIP3 consistent time series whenever the reference data was based on the older generation of SRES and ENSEMBLES scenarios. The quality of the sea level model calibration is therefore not affected by the warmer MAGICC air temperature response. Our primary 2100 SLR projections are based on a MAGICC ensemble that is constrained by CMIP5 global mean *tas*. These projections can therefore be directly compared to recent IPCC estimates. For our 2300 projections, we run MAGICC

in the historically-constrained, probabilistic setup described above. The resulting MAGICC air temperature responses mostly reflect the available CMIP5 reference data, although they show a shorter response time scale (see Figure 5). These differences to CMIP5 translate into the corresponding SLR projections due to the strong air temperature dependence of the sea level model. Hence, the MAGICC sea level module will only be able to provide fully CMIP5 consistent SLR responses for 2300 once the remaining components of the MAGICC model have been updated.

Sea level emulators cannot substitute comprehensive process-based sea level projections. Emulated sea level projections reflect, independently, the reference responses for each individual sea level component, assuming that the implemented parameterizations fully capture the process-based simulations. Due to their computational efficiency, sea level emulators have the ability to map a wider range of scenario uncertainties for each sea level contribution than the process-based models used for the calibration. This defines a key strength of the MAGICC sea level model and makes it a useful tool to assess sea level

trajectories for scenarios that are not covered by the large comprehensive models. It is well known that model uncertainties differ substantially for the individual sea level components (Church et al., 2013a). In 2300, the three largest model response uncertainties captured by the MAGICC sea level model for RCP8.5 are the Greenland SMB component with 66% range estimates of 1.01 m to 3.33 m, the thermal expansion component with a 66% range of 1.03 m to 2.45 m, and the Antarctic SID component with 0.73 m to 2.12 m. Emulators, as presented here, can only cover the uncertainty ranges that are reflected

in the emulated process-based models. Even though there have been substantial advances in process understanding over the





last years, the physical representation of some sea level contributions remains incomplete. The Antarctic ice sheet response, for example, could be subject to more rapid, non-linear dynamics that are not captured by current process-based projections. Only recently, DeConto and Pollard (2016) have revised potential future Antarctic contributions to global sea level based on indicators from paleoclimatic archives. For RCP8.5, 2100 contributions of around 1m are suggested from Antarctica alone,

with 2300 contributions reaching up to around 10 m. These numbers illustrate the need to handle long-term SLR projections with particular care and provide the corresponding methodological caveats.

The MAGICC sea level model assesses long-term global SLR trajectories by synthesizing available process-based projections for the individual sea level drivers and applying them to the available set of RCP scenarios and their extensions until 2300. The current version shows 2100 estimates that are well within the range of the latest IPCC assessment (see Figure 4).

The structure of the emulator makes the MAGICC sea level model a computationally much more efficient tool compared to the comprehensive and complex process-based models. The calibration routines for the individual components have been flexibly designed to allow for timely updates whenever new robust modeling results become available. The presented MAGICC sea level model, together with the MAGICC ocean model update, are new elements of MAGICC model version 6 (Meinshausen et al., 2011a). The implementation of the new sea level model initiates the development of MAGICC model version 7 to

comprehensively emulate CMIP5 projections. The full potential of the MAGICC sea level model will be unlocked once this MAGICC model upgrade has been completed.

## 5   Model code and data availabilty

The Fortran code of the MAGICC sea level model together with the respective documentation is available here:
https://gitlab.com/anauels/MAGICC_SLR_model.

Output data from the conducted experiments is available on request. The calibration data is either freely available from the CMIP5 database http://cmip-pcmdi.llnl.gov/ or has to be requested from the authors of the corresponding reference studies.

*Author contributions.*  A. Nauels developed and calibrated the sea level model with support from M. Meinshausen, M. Mengel, and K. Lorbacher. A. Nauels conducted the experiments and drafted the manuscript. All authors contributed to the text and declare that they have no conflict of interest.

*Acknowledgements.*  We acknowledge the World Climate Research Programme's Working Group on Coupled Modelling, which is responsible for CMIP, and we thank the climate modeling groups for producing and making available their model output (applied CMIP5 models are listed in Table 1 of this paper, see also http://cmip-pcmdi.llnl.gov/cmip5/docs/CMIP5_modeling_groups.pdf). For CMIP the U.S. Department of Energy's Program for Climate Model Diagnosis and Intercomparison provides coordinating support and led development of software infrastructure in partnership with the Global Organization for Earth System Science Portals. We would especially like to thank B. Marzeion,

X. Fettweis, F. Nick, S. Ligtenberg, A. Levermann, and Y. Wada for providing the calibration data used in this study. M. Meinshausen receives





the Australian Research Council (ARC) Future Fellowship Grant FT130100809. T. M. L. Wigley is supported by the Australian Research Council under Discovery Grant DP130103261.





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



**Table 1.** MAGICC ocean model calibration results with optimal sets of ocean and thermal expansion calibration parameters for the available CMIP5 models. Calibration parameters are introduced in section 2.6. Goodness-Of-Fit (GOF) results are given as weighted Residual Sum of Squares (RSS) divided by the number of calibrated model years (weight potential ocean temperature [K]: 10, weight thermal expansion [mm]: 0.001). The optimal set for the mean response of the calibration data is given at the bottom of the table.

| Model | $K_z$ | $\frac{dK_{z_{top}}}{dT}$ | $\eta$ | $\gamma$ | $\beta$ | $w_0$ | $\frac{\Delta w_t}{w_t}$ | $T_{w_t}$ | $\phi$ | GOF |
|---|---|---|---|---|---|---|---|---|---|---|
| ACCESS1.0 | 0.0114 | -1.2808 | 3.2171 | 0.2764 | 0.0100 | 10.000 | 0.6905 | 11.766 | 1.2105 | 0.10 |
| ACCESS1.3 | 0.3245 | -0.0711 | 2.1479 | 0.1972 | 0.0111 | 10.000 | 0.9673 | 15.790 | 1.2043 | 0.12 |
| BCC-CSM1.1 | 1.2371 | 0.1152 | 1.9303 | 0.3281 | 0.1485 | 0.4889 | 0.1947 | 19.999 | 1.0275 | 0.13 |
| BCC-CSM1.1M | 0.5492 | 0.4344 | 2.1254 | 0.1868 | 0.3983 | 0.0100 | 0.1718 | 9.6084 | 1.0243 | 0.12 |
| BNU-ESM | 0.0471 | 0.7137 | 2.2079 | 0.4412 | 1.0000 | 1.9698 | 0.0044 | 19.623 | 1.2714 | 0.28 |
| CanESM2 | 0.4651 | 0.4465 | 2.8877 | 0.2828 | 0.4648 | 0.6509 | 0.5654 | 2.9600 | 1.0777 | 0.13 |
| CCSM4 | 1.1052 | 0.1013 | 1.4828 | 0.4402 | 0.5952 | 1.1499 | 1.0000 | 5.9815 | 1.0342 | 0.14 |
| CESM1-BGC | 0.0214 | 0.3504 | 1.5848 | 0.4561 | 0.6230 | 2.5047 | 0.5976 | 3.7201 | 1.0495 | 0.05 |
| CESM1-CAM5 | 0.0100 | 0.7161 | 1.7729 | 0.5131 | 0.3377 | 0.1160 | 0.3818 | 8.7073 | 1.1201 | 0.15 |
| CMCC-CESM | 0.1876 | -1.5000 | 1.2706 | 0.3567 | 0.8683 | 1.5076 | 0.5915 | 1.5371 | 1.1077 | 0.09 |
| CMCC-CM | 0.0100 | 0.4864 | 3.3366 | 0.2042 | 0.7504 | 1.1562 | 0.1229 | 7.7459 | 1.0242 | 0.15 |
| CMCC-CMS | 0.1271 | 0.0441 | 1.9926 | 0.3231 | 0.0113 | 2.8231 | 0.3850 | 3.5972 | 1.2412 | 0.04 |
| CNRM-CM5 | 0.9357 | 0.0116 | 2.9140 | 0.2127 | 0.0487 | 0.0101 | 0.1805 | 0.0021 | 1.0576 | 0.21 |
| CNRM-CM5-2 | 1.7254 | 0.6964 | 1.7609 | 0.5229 | 0.1296 | 9.9949 | 0.0783 | 17.826 | 1.3039 | 0.05 |
| CSIRO-Mk3.6.0 | 1.8315 | -0.0090 | 3.9397 | 0.1395 | 0.0397 | 0.0102 | 0.3776 | 19.999 | 1.0792 | 0.47 |
| EC-EARTH | 2.1902 | -0.3179 | 3.9198 | 0.2118 | 0.0540 | 4.3349 | 0.3310 | 19.999 | 1.0894 | 0.07 |
| GFDL-CM3 | 0.0100 | 0.1233 | 1.8490 | 0.3985 | 0.0382 | 8.4294 | 0.7954 | 15.640 | 1.2551 | 0.73 |
| GFDL-ESM2G | 0.4746 | -0.9556 | 1.8659 | 0.2713 | 1.0000 | 5.3175 | 0.3200 | 4.8205 | 1.2693 | 0.25 |
| GFDL-ESM2M | 1.9860 | 1.0000 | 1.3911 | 0.5918 | 0.3478 | 1.4574 | 0.0010 | 17.181 | 1.1661 | 0.13 |
| GISS-E2-H | 1.0904 | 0.4033 | 1.5139 | 0.5116 | 0.4742 | 1.6710 | 0.5888 | 5.9413 | 1.1010 | 0.09 |
| GISS-E2-HCC | 0.0223 | -1.1885 | 1.4176 | 0.3284 | 0.0514 | 10.000 | 0.9909 | 17.001 | 1.2042 | 0.09 |
| GISS-E2-R | 1.7180 | 0.7485 | 1.2225 | 0.7365 | 0.0566 | 2.4337 | 0.4215 | 1.9195 | 1.0751 | 0.19 |
| GISS-E2-RCC | 5.0000 | -1.0798 | 1.0847 | 0.7858 | 0.1896 | 7.2667 | 0.2946 | 6.3640 | 1.1186 | 0.13 |
| HadGEM2-CC | 0.1186 | -1.4379 | 2.8436 | 0.3196 | 0.2976 | 6.5513 | 0.5173 | 8.0253 | 1.1712 | 0.07 |
| HadGEM2-ES | 0.7873 | -0.0167 | 3.0799 | 0.2488 | 0.3899 | 3.2808 | 0.3344 | 6.9478 | 1.1255 | 0.54 |
| IPSL-CM5A-LR | 1.0325 | -0.0163 | 2.4490 | 0.1991 | 0.3023 | 1.5790 | 0.7269 | 19.999 | 1.0724 | 0.11 |
| IPSL-CM5A-MR | 0.7161 | 0.1700 | 2.8728 | 0.1549 | 0.9991 | 0.3723 | 0.0433 | 6.9285 | 1.0860 | 0.09 |
| IPSL-CM5B-LR | 0.7926 | 0.5048 | 1.9514 | 0.4509 | 0.0115 | 6.559 | 0.0010 | 7.6319 | 1.3134 | 0.23 |
| MIROC5 | 0.0118 | 0.9417 | 2.8825 | 0.2336 | 0.0100 | 7.4846 | 0.8527 | 12.335 | 1.1908 | 0.19 |
| MIROC-ESM | 0.0100 | 0.4039 | 1.5715 | 0.6060 | 0.6477 | 1.9151 | 0.6013 | 3.2088 | 1.0876 | 0.18 |
| MIROC-ESM-CHEM | 0.0100 | -0.0039 | 4.9735 | 0.0972 | 0.0356 | 10.000 | 0.8157 | 12.410 | 1.1396 | 0.06 |
| MPI-ESM-LR | 1.4331 | -1.5000 | 3.1216 | 0.1796 | 0.0764 | 8.8280 | 0.4413 | 6.6323 | 1.2689 | 0.40 |
| MPI-ESM-MR | 0.8171 | -0.8556 | 2.1675 | 0.3558 | 0.0101 | 10.000 | 0.9358 | 14.076 | 1.1382 | 0.06 |
| MPI-ESM-P | 2.4551 | -1.4726 | 3.5690 | 0.1246 | 0.0431 | 9.3941 | 0.5070 | 6.7223 | 1.2246 | 0.07 |
| MRI-CGCM3 | 1.0157 | 0.8451 | 1.9347 | 0.3882 | 0.1549 | 9.9962 | 0.0010 | 19.411 | 1.1115 | 0.05 |
| NorESM1-M | 2.0596 | -1.3630 | 4.9675 | 0.1071 | 1.0000 | 5.2985 | 0.9999 | 16.449 | 1.1825 | 0.14 |
| NorESM1-ME | 1.5726 | 0.9637 | 1.7980 | 0.3792 | 0.4383 | 6.2192 | 0.1148 | 20.000 | 1.1160 | 0.08 |
| Mean | 0.2505 | -1.3057 | 2.4741 | 0.2724 | 0.4644 | 9.9738 | 0.8179 | 12.368 | 0.90782 | 0.05 |





**Table 2.** Glacier sea level component calibration results with parameter sets for the available CMIP5 models. Calibration parameters are introduced in section 2.6. GOF is given as weighted RSS divided by the number of calibrated model years (weight glacier SLE contribution [mm]: 1). The optimal set for the mean response of the calibration data is given at the bottom of the table.

| Model | $\kappa$ | $\nu$ | GOF |
|---|---|---|---|
| BCC-CSM1.1 | 0.0114 | 0.2779 | 58.99 |
| CanESM2 | 0.0102 | 0.0553 | 29.47 |
| CCSM4 | 0.0109 | 0.1432 | 12.92 |
| CNRM-CM5 | 0.0089 | 0.2678 | 175.9 |
| CSIRO-Mk3.6.0 | 0.0094 | 0.2431 | 146.5 |
| GFDL-CM3 | 0.0108 | 0.2043 | 19.22 |
| GISS-E2-R | 0.0104 | 0.1257 | 32.39 |
| HadGEM2-ES | 0.0097 | 0.2930 | 97.24 |
| IPSL-CM5A-LR | 0.0084 | 0.2512 | 28.87 |
| MIROC5 | 0.0116 | 0.0747 | 18.70 |
| MIROC-ESM | 0.0089 | 0.1279 | 60.51 |
| MPI-ESM-LR | 0.0099 | 0.2664 | 38.37 |
| MRI-CGCM3 | 0.0071 | 0.1473 | 20.28 |
| NorESM1-M | 0.0095 | 0.1129 | 36.85 |
| Mean | 0.0094 | 0.0680 | 12.79 |



**Table 3.** Greenland SMB sea level component calibration results with optimal parameter sets for the available CMIP5 models. Calibration parameters are introduced in section 2.6. GOF is given as weighted RSS divided by the number of calibrated model years (weight Greenland SMB SLE contribution [mm]: 1). The optimal set for the mean response of the calibration data is given at the bottom of the table.

| Model | $\upsilon$ | $\chi$ | $\varphi$ | GOF |
|---|---|---|---|---|
| ACCESS1.0 | 0.1527 | 0.9636 | 3.3613 | 0.75 |
| ACCESS1.3 | 0.2310 | 0.9979 | 4.3640 | 0.38 |
| BCC-CSM1.1 | 0.0506 | 0.3298 | 2.6337 | 0.58 |
| BNU-ESM | 0.0496 | 0.0000 | 2.4554 | 1.47 |
| CanESM2 | 0.064 | 0.4813 | 2.7418 | 2.15 |
| CCSM4 | 0.0182 | 0.0000 | 2.7886 | 1.15 |
| CESM1-BGC | 0.0348 | 0.0000 | 2.3174 | 1.12 |
| CMCC-CM | 0.0595 | 0.0000 | 2.1369 | 1.41 |
| CNRM-CM5 | 0.0690 | 0.0000 | 2.0173 | 0.33 |
| CSIRO-Mk3.6.0 | 0.1620 | 0.8580 | 2.5187 | 0.55 |
| GFDL-CM3 | 0.1889 | 0.1542 | 2.0905 | 0.82 |
| GFDL-ESM2M | 0.0767 | 0.0000 | 2.3222 | 0.96 |
| GISS-E2-R | 0.0896 | 0.0000 | 2.2902 | 0.31 |
| HadGEM2-CC | 0.1263 | 0.8498 | 2.7489 | 0.87 |
| HadGEM2-ES | 0.1133 | 0.4669 | 2.0736 | 0.65 |
| IPSL-CM5A-LR | 0.1480 | 0.4805 | 2.0264 | 0.22 |
| IPSL-CM5A-MR | 0.0822 | 0.0000 | 2.0509 | 0.57 |
| IPSL-CM5B-LR | 0.0260 | 0.0000 | 2.9233 | 1.08 |
| MIROC5 | 0.1844 | 0.0000 | 1.9012 | 1.03 |
| MIROC-ESM-CHEM | 0.1667 | 0.4959 | 2.2633 | 1.41 |
| MIROC-ESM | 0.0976 | 0.0000 | 2.2699 | 0.76 |
| MPI-ESM-LR | 0.0407 | 0.0000 | 2.5751 | 1.20 |
| MRI-CGCM3 | 0.0418 | 0.0000 | 2.56 | 0.57 |
| NorESM1-M | 0.0762 | 0.0000 | 2.1049 | 0.57 |
| Mean | 0.1176 | 0.0000 | 1.9990 | 0.37 |





**Table 4.** Greenland SID sea level component calibration results with optimal parameter sets for the minimum and maximum cases introduced by Nick et al. (2013). Calibration parameters are introduced in section 2.6. GOF is given as weighted RSS divided by the number of calibrated model years (weight Greenland SID SLE contribution [mm]: 1).

| Case | $\varrho$ | $\epsilon$ | $\zeta$ | $GIS_{max}^{outlet}$ [mm] | GOF |
|------|-----------|------------|---------|--------------------------|-----|
| min | 4.9962e-07 | 1.5672 | 4.3811e+03 | 40.46 | 0.23 |
| max | 1.1119e-05 | 1.1529 | 2.2352e+02 | 53.67 | 0.48 |

**Table 5.** Antarctic SMB sea level component calibration results with optimal parameter sets for the CMIP3 models ECHAM5 and HadCM3. Calibration parameters are introduced in section 2.6. GOF is given as weighted RSS divided by the number of calibrated model years (weight Antarctic SMB SLE contribution [mm]: 1). The optimal set for the mean response of the calibration data is given at the bottom of the table.

| Model | $\xi$ | $\rho$ | $\sigma$ | GOF |
|-------|-------|--------|----------|-----|
| ECHAM5 | 0.1280 | -0.4244 | -0.7819 | 0.70 |
| HadCM3 | -0.2900 | -0.1987 | 0.4646 | 9.59 |
| Mean | -0.1208 | 0.0000 | 1.5232 | 0.70 |



**Table 6.** 2081-2100 median values and 66% ranges for global SLR projections in meters resolved by sea level components for the four RCP scenarios. Estimates are provided based on the CMIP5 consistent MAGICC setup. IPCC median projections and likely ranges are given as a reference.

| 2081-2100 | | RCP2.6 | RCP4.5 | RCP6.0 | RCP8.5 |
|---|---|---|---|---|---|
| Total | MAGICC | 0.43 [0.35 to 0.54] | 0.51 [0.42 to 0.61] | 0.51 [0.42 to 0.63] | 0.67 [0.57 to 0.82] |
| | IPCC | 0.40 [0.26 to 0.55] | 0.47 [0.32 to 0.63] | 0.48 [0.33 to 0.63] | 0.63 [0.45 to 0.82] |
| Thermal Expansion | MAGICC | 0.12 [0.08 to 0.17] | 0.16 [0.12 to 0.21] | 0.17 [0.12 to 0.22] | 0.25 [0.20 to 0.33] |
| | IPCC | 0.14 [0.10 to 0.18] | 0.19 [0.14 to 0.23] | 0.19 [0.15 to 0.24] | 0.27 [0.21 to 0.33] |
| Glaciers | MAGICC | 0.11 [0.09 to 0.13] | 0.12 [0.10 to 0.14] | 0.12 [0.10 to 0.14] | 0.14 [0.12 to 0.16] |
| | IPCC | 0.10 [0.04 to 0.16] | 0.12 [0.06 to 0.19] | 0.12 [0.06 to 0.19] | 0.16 [0.09 to 0.23] |
| Greenland SMB | MAGICC | 0.02 [0.01 to 0.04] | 0.03 [0.02 to 0.05] | 0.03 [0.02 to 0.05] | 0.06 [0.04 to 0.09] |
| | IPCC | 0.03 [0.01 to 0.07] | 0.04 [0.01 to 0.09] | 0.04 [0.01 to 0.09] | 0.07 [0.03 to 0.16] |
| Greenland SID | MAGICC | 0.04 [0.03 to 0.05] | 0.04 [0.04 to 0.05] | 0.04 [0.04 to 0.05] | 0.05 [0.04 to 0.05] |
| | IPCC | 0.04 [0.01 to 0.06] | 0.04 [0.01 to 0.06] | 0.04 [0.01 to 0.06] | 0.05 [0.02 to 0.07] |
| Antarctica SMB | MAGICC | -0.02 [-0.03 to -0.01] | -0.02 [-0.03 to 0.02] | -0.02 [-0.03 to 0.02] | -0.03 [-0.05 to -0.03] |
| | IPCC | -0.02 [-0.04 to -0.00] | -0.02 [-0.05 to -0.01] | -0.02 [-0.05 to -0.01] | -0.04 [-0.07 to -0.01] |
| Antarctica SID | MAGICC | 0.07 [0.04 to 0.15] | 0.09 [0.05 to 0.17] | 0.09 [0.05 to 0.17] | 0.11 [0.06 to 0.22] |
| | IPCC | 0.07 [-0.01 to 0.16] | 0.07 [-0.01 to 0.16] | 0.07 [-0.01 to 0.16] | 0.07 [-0.01 to 0.16] |
| Land water storage | MAGICC | 0.05 [0.04 to 0.06] | 0.05 [0.04 to 0.06] | 0.05 [0.04 to 0.06] | 0.05 [0.04 to 0.06] |
| | IPCC | 0.04 [-0.01 to 0.09] | 0.04 [-0.01 to 0.09] | 0.04 [-0.01 to 0.09] | 0.04 [-0.01 to 0.09] |





**Table 7.** 2100 and 2300 median values as well as 66% ranges for total global SLR projections based on the MAGICC CMIP5 and MAGICC
PROB experimental designs. IPCC median projections and likely ranges are given as a reference.

|  |  | 2100 | 2300 |
| --- | --- | --- | --- |
| | MAGICC CMIP5 | 0.47 [0.38 to 0.59] | - |
| RCP2.6 | MAGICC PROB | 0.46 [0.36 to 0.60] | 0.97 [0.76 to 1.35] |
| | IPCC | 0.44 [0.28 to 0.61] | - |
| | MAGICC CMIP5 | 0.56 [0.47 to 0.68] | - |
| RCP4.5 | MAGICC PROB | 0.58 [0.46 to 0.76] | 1.65 [1.23 to 2.37] |
| | IPCC | 0.53 [0.36 to 0.71] | - |
| | MAGICC CMIP5 | 0.58 [0.47 to 0.72] | - |
| RCP6.0 | MAGICC PROB | 0.63 [0.50 to 0.82] | 2.32 [1.67 to 3.41] |
| | IPCC | 0.55 [0.38 to 0.73] | - |
| | MAGICC CMIP5 | 0.79 [0.66 to 0.96] | - |
| RCP8.5 | MAGICC PROB | 0.87 [0.69 to 1.17] | 5.11 [3.63 to 7.55] |
| | IPCC | 0.74 [0.52 to 0.98] | - |





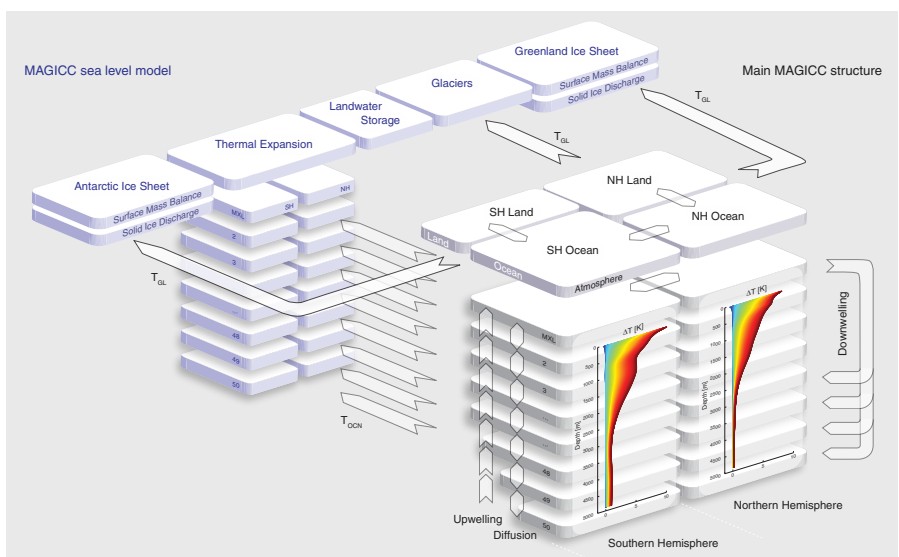

**Figure 1.** Schematic of the MAGICC sea level model structure and the driving MAGICC hemispheric upwelling-diffusion energy balance core. Heat is transported through the oceans by downwelling and corresponding layer entrainment, upwelling, diffusion, and the exchange between the hemispheres. Ocean mixed layer is denoted MXL, depth-dependent ocean areas are shown by smaller ocean layers towards the ocean bottom. Illustrative potential ocean temperature warming profiles that feed into the layer-dependent thermal expansion module are sketched for both hemispheres. Ocean and air temperature fluxes ($T_{OCN}$, $T_{GL}$) relevant for the sea level model as well as other major energy fluxes are shown as arrows. Figure adapted from Meinshausen et al. (2011a).





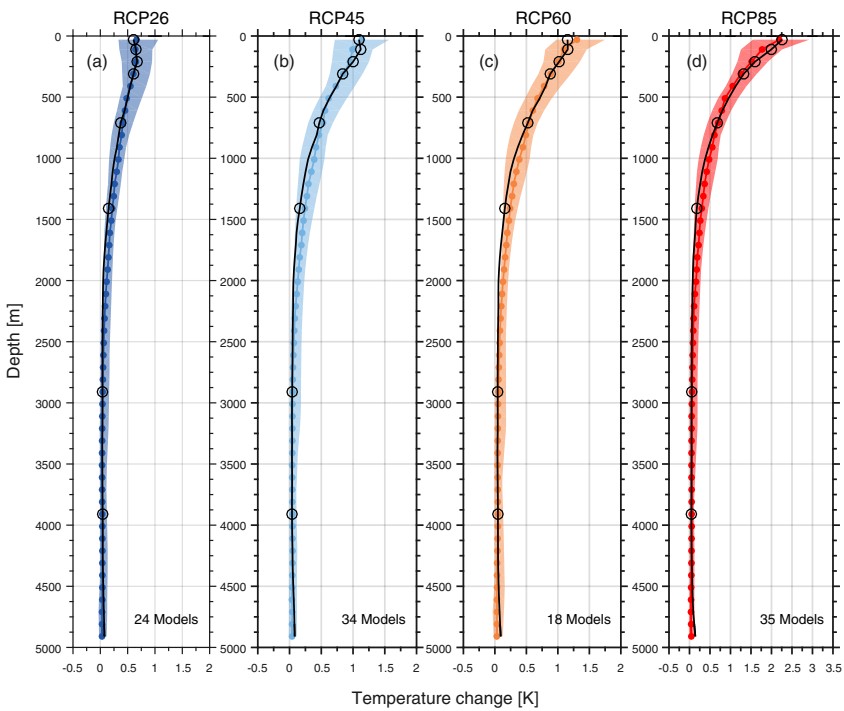

**Figure 2.** Potential ocean temperature depth profiles for MAGICC and reference CMIP5 warming under RCP2.6, RCP4.5, RCP6.0 and RCP8.5 scenarios, 2081-2100 anomalies with respect to 1986-2005. Interpolated CMIP5 90% model ranges and corresponding median profiles are shown in colors, with circles indicating the individual MAGICC ocean layers. MAGICC median ocean warming profiles given as black lines with open circles indicating selected layers for ocean calibration.





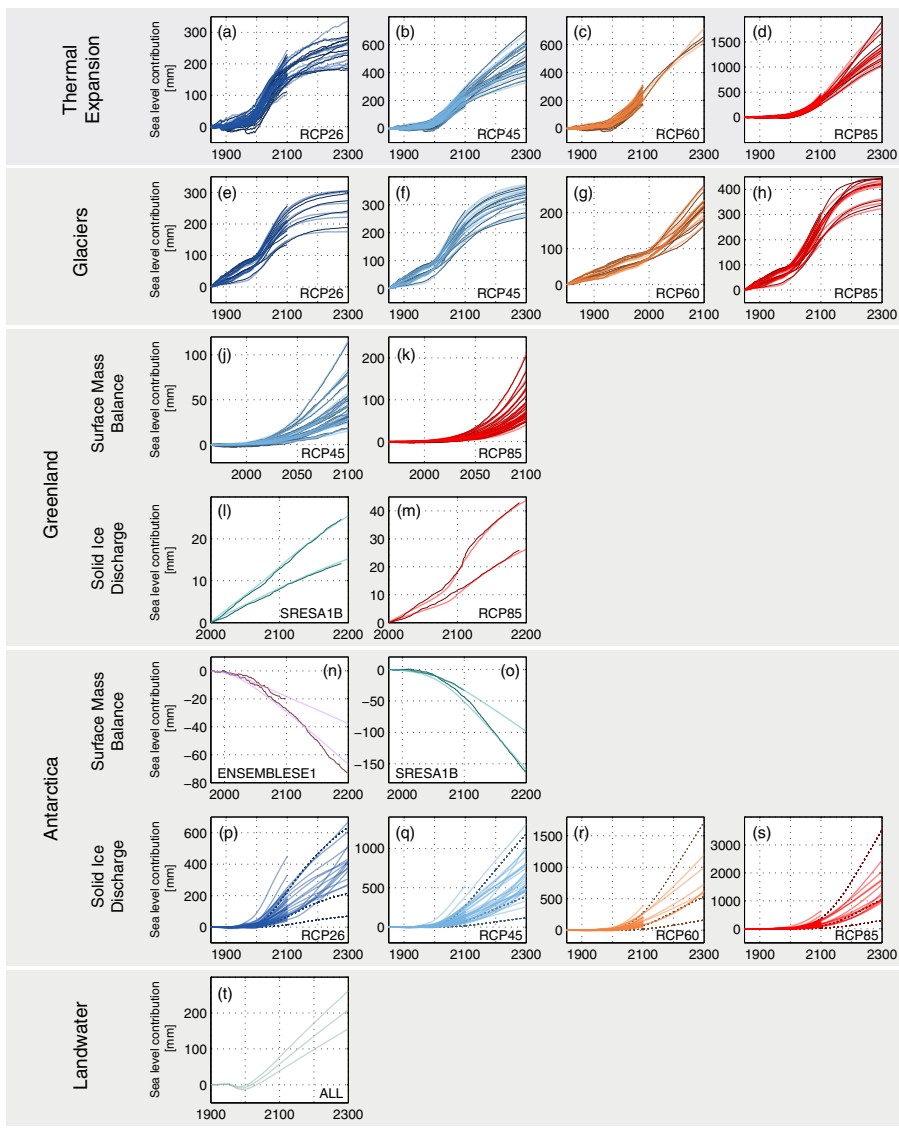

**Figure 3.** MAGICC sea level model calibration results for thermal expansion (a-d) , global glaciers (e-h), Greenland surface mass balance (j-k) and solid ice discharge (l-m), Antarctic surface mass mass balance (n-o) and solid ice discharge (p-s), as well as land water (t). The panels show scenario-specific calibrated MAGICC sea level responses as coloured lines, with underlying reference data as thin dark lines. Antarctic solid ice discharge reference 90% range plus corresponding median are provided as thin dashed lines. Climate-independent land water projections are identical to the reference data until 2100 (see Section 2.5).





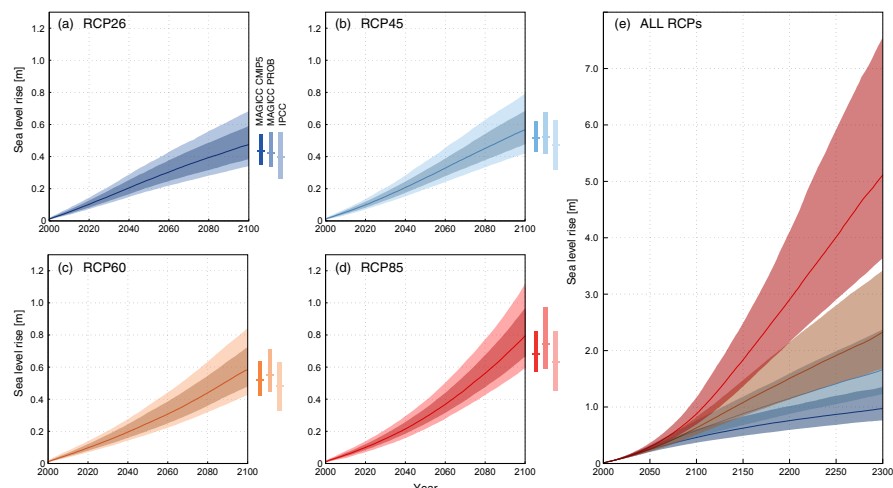

**Figure 4.** Global sea level projections until 2100 based on CMIP5 constrained MAGICC runs as anomalies relative to 1986-2005 in panels (a) to (d). 90% ensemble range in light colors, 66% ensemble range in darker colors, median as solid line. 2081-2100 anomalies with respect to 1986-2005 as vertical bars for CMIP5 constrained MAGICC setup (MAGICC CMIP5), historically-constrained probabilistic MAGICC setup (MAGICC PROB), and IPCC reference projections. 2300 sea level projections in panel (e) are showing 66% ranges for all RCP extensions based on MAGICC PROB; median estimates as solid lines.





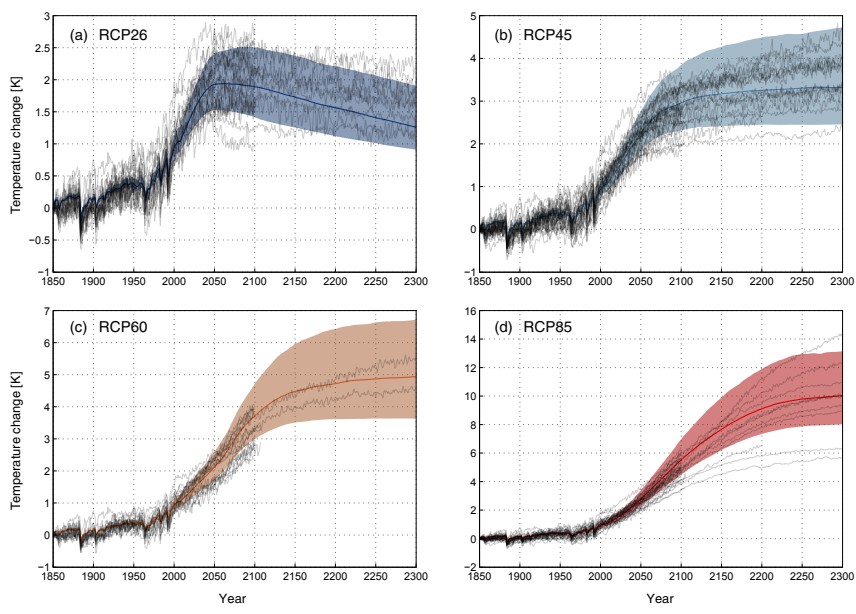

**Figure 5.** Global mean *tas* projections until 2300 for all RCP extensions based on the historically-constrained probabilistic MAGICC setup; 66% ensemble ranges with median estimates as solid lines. Available global CMIP5 *tas* reference time series are shown as thin black lines. All temperature projections are given relative to 1850.



**Appendix A: 2300 SLR projections resolved by the indiviual sea level components for all RCP scenarios**





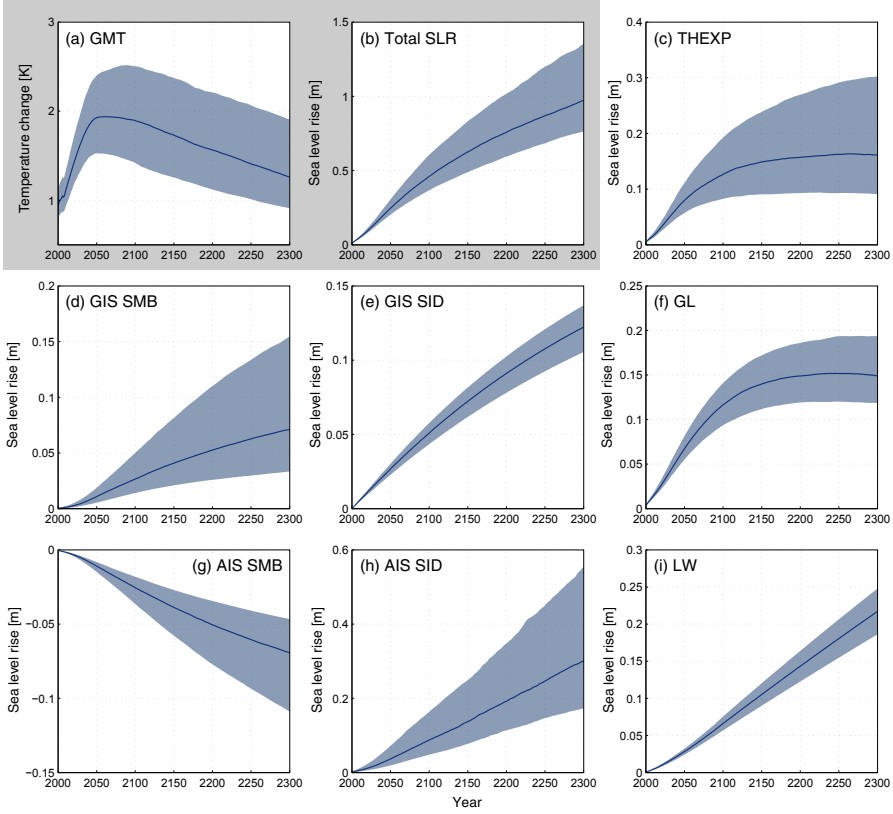

**Figure A.1.** 2300 SLR projections resolved by the indiviual MAGICC sea level components for RCP2.6 in meters. We show median estimates and 66% ranges for MAGICC GMT output relative to pre-industrial in panel (a), total SLR in panel (b), thermal expansion (THEXP) in panel (c), Greenland Ice Sheet (GIS) SMB and SID in panels (d) and (e), global glacier (GL) in panel (f), Antarctic ice sheet (AIS) SMB and SID in panels (g) and (h), and land water LW in panel (i). All sea level contributions are provided relative to 1986-2005.





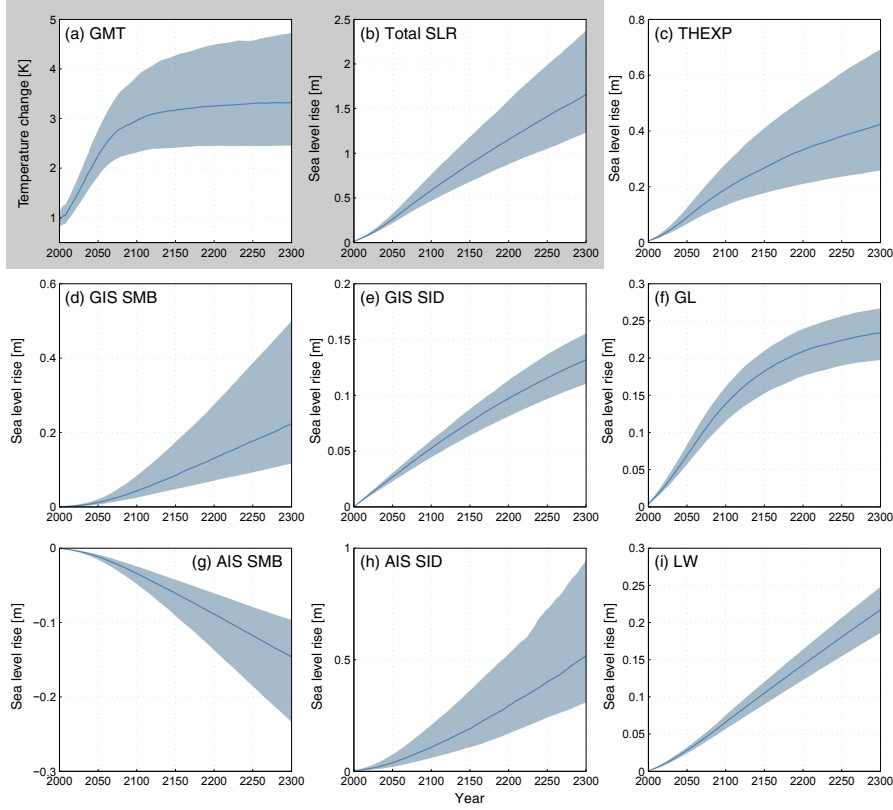

**Figure A.2.** 2300 SLR projections resolved by the indiviual MAGICC sea level components for RCP4.5 in meters. We show median estimates and 66% ranges for MAGICC GMT output relative to pre-industrial in panel (a), total SLR in panel (b), thermal expansion (THEXP) in panel (c), Greenland Ice Sheet (GIS) SMB and SID in panels (d) and (e), global glacier (GL) in panel (f), Antarctic ice sheet (AIS) SMB and SID in panels (g) and (h), and land water LW in panel (i). All sea level contributions are provided relative to 1986-2005.



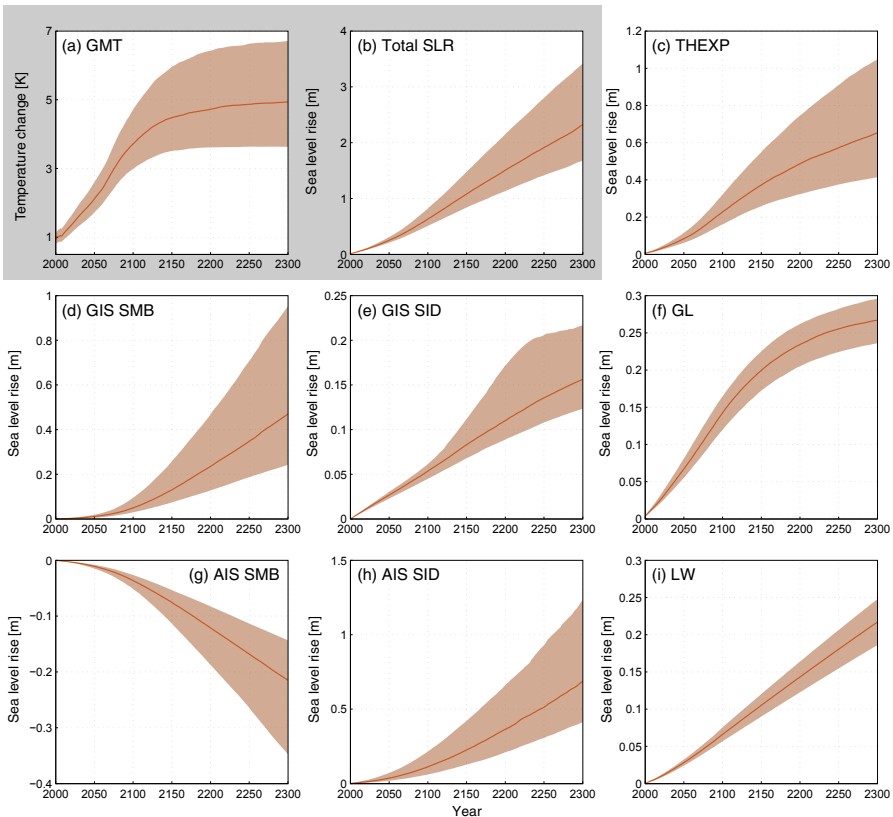

**Figure A.3.** 2300 SLR projections resolved by the indiviual MAGICC sea level components for RCP6.0 in meters. We show median estimates and 66% ranges for MAGICC GMT output relative to pre-industrial in panel (a), total SLR in panel (b), thermal expansion (THEXP) in panel (c), Greenland Ice Sheet (GIS) SMB and SID in panels (d) and (e), global glacier (GL) in panel (f), Antarctic ice sheet (AIS) SMB and SID in panels (g) and (h), and land water LW in panel (i). All sea level contributions are provided relative to 1986-2005.





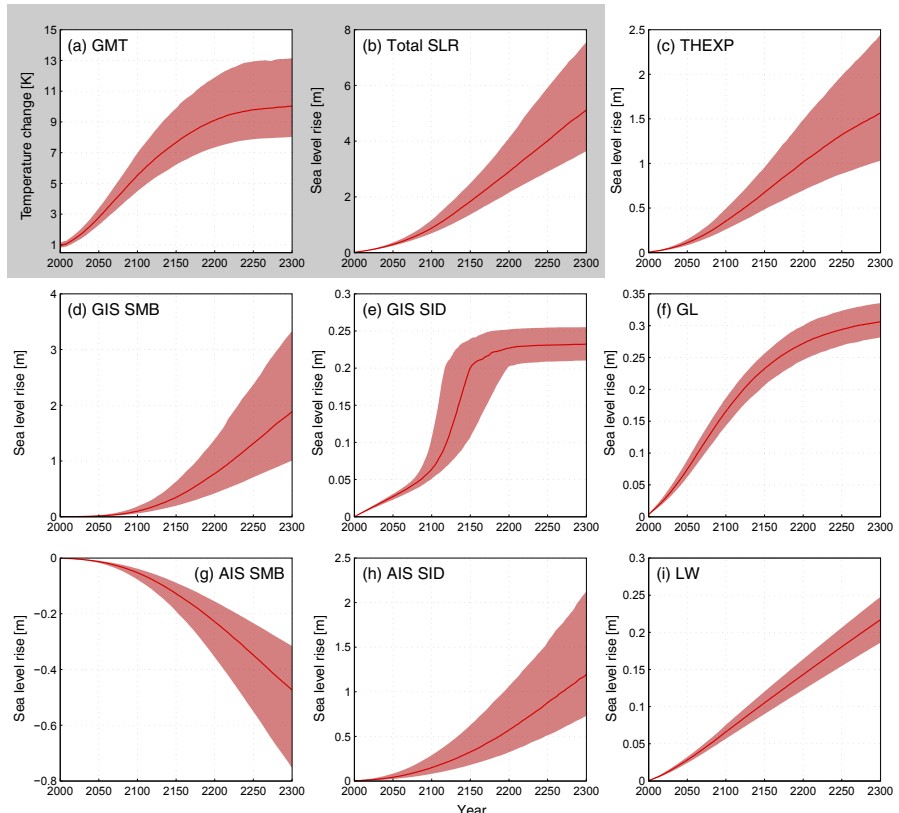

**Figure A.4.** 2300 SLR projections resolved by the indiviual MAGICC sea level components for RCP8.5 in meters. We show median estimates and 66% ranges for MAGICC GMT output relative to pre-industrial in panel (a), total SLR in panel (b), thermal expansion (THEXP) in panel (c), Greenland Ice Sheet (GIS) SMB and SID in panels (d) and (e), global glacier (GL) in panel (f), Antarctic ice sheet (AIS) SMB and SID in panels (g) and (h), and land water LW in panel (i). All sea level contributions are provided relative to 1986-2005.