# Peer review of "Synthesizing long-term sea level rise projections - the MAGICC sea level model v2.0"

_Geoscientific Model Development, 2016_

## Short Comment (SC1) · 24 Oct 2016

Dear authors,

In my role as Executive editor of GMD, I would like to bring to your attention our Editorial version 1.1:

http://www.geosci-model-dev.net/8/3487/2015/gmd-8-3487-2015.html

This highlights some requirements of papers published in GMD, which is also available on the GMD website in the 'Manuscript Types' section:

http://www.geoscientific-model-development.net/submission/manuscript_types.html

In particular, please note that for your paper, the following requirement has not been met in the Discussions paper:

[Figure]

- "The main paper must give the model name and version number (or other unique identifier) in the title."

Please add a version number for MAGICC in the title upon your revised submission to GMD.

Additionally, please note that the Code and Data Availability Section is not a numbered Section.

Yours,

Astrid Kerkweg
* * *

---

## Referee Comment (RC1) · Anonymous Referee #1 · 10 Feb 2017

In this paper, the authors describe the first version of the MAGICC sea level model, a set of sea-level emulators linked to versions 6 and 7 of the MAGICC climate model. They provide a clear description of the algorithms for the different sea-level components and of the model calibration. Accordingly, I believe the paper is worth publishing and have only minor comments regarding the paper itself.

I suggest the authors add some discussion placing the MAGICC sea level model in the context of other similar tools, such as the BRICK model, also currently in review at GMD (http://dx.doi.org/10.5194/gmd-2016-303). A comparison of the model results to that of other simple sea-level models (e.g., Kopp et al., 2016, and Mengel et al., 2016) under similar forcing would be helpful – prima facie, the projections for 2100 seem to align well with these simpler models.

[Figure]

Line 28: I believe the citation here should be to Levermann et al. (2014), not Levermann et al. (2013).

Line 30: Note that an early semiempirical model was introduced by Gornitz et al. (1982) twenty five years earlier. Gornitz, V., S. Lebedeff, and J. Hansen, 1982: Global sea level trend in the past century. Science, 215, 1611–1614, doi:10.1126/science.215.4540.1611.

I also note that the code for the sea level model, while available at the gitlab link provided, does not run without the MAGICC model, for which code is not available. I further note that GMD policy states: "If the authors cannot or do not wish to make the code and/or data public (e.g. copyright or licensing restrictions), the reasons must be clearly stated. Note that, for the purpose of the review, the code and/or data must still be made available to the editor. Access must also be granted to the reviewers whilst preserving their anonymity, if this is legally possible." I am not sure whether the current code availability – dependent upon code that was not available for the review process, and for whose non-availability no explanation was provided – satisfies this policy.

---

## Referee Comment (RC2) · Anonymous Referee #2 · 24 Feb 2017

General Comments:

The ms tackles an important and nontrivial point and makes several new contributions. The authors are to be applauded for the efforts and providing the new insights. Alas, the current ms suffers from several (in my view severe, but pretty easy to fix) shortcomings that need to be mitigated to enable a careful review and to hopefully (and eventually) rise to the level of quality one expects from papers in GMD. These specific points are discussed below. The ms requires in my view a careful and responsive revision and re-review. I would be happy to look at this ms again.

Specific Points:

The ms does, in my view, not follow the data and code policy of GMD. As a reminder, here is the link describing the policy.

http://www.geoscientific-model-development.net/about/code_and_data_policy.html
These rules state: "Preferably, this section should contain the instructions for obtaining the model code and/or data, either from the supplement or from an archive with a digital object identifier (DOI). Suitable repositories can be found at the Registry of Research Data Repositories, e.g. ZENODO for model code. After the paper is accepted, a link to the GMD paper should be added to the metadata of the archive." What is available, as far as I can tell, is code for the sea-level rise model / module. However, it will be extremely tough to reproduce the results in the paper, due to several issues. For example, the interface to call this module is not well documented, nor are the data used to drive and calibrate the model provided. The web-page the ms links to states: "The Fortran code of the model as well as required additional MAGICC input files can be downloaded as a zip-file at the top right of this page. Please note that the MAGICC sea level model is not a stand-alone tool. It is fully integrated into the simple carbon-cycle climate model MAGICC and cannot be run separately. The MAGICC model is described in Meinshausen et al. (2011). For a compiled MAGICC version including the sea level model, please contact alexander.nauels@climate-enegy-college.org. MAGICC is licensed under a Creative Commons Attribution-NonCommercial-ShareAlike 3.0 Unported License." Does this mean that the source code for the model needed to run the sea-level model is not accessible? Why is only a compiled version distributed? Given this situation, I would assign a very low score for reproducibility (see the discussion by Haas in the recent Risk Analysis volume (Risk Analysis, Vol. 36, No. 10, 2016, DOI: 10.1111/risa.12730). Many other studies have abided by the nice standards outlined in GMD and a publication in GMD should clearly abide by these standards. This is a decision for the editor. I cannot carefully review the ms without the ability to see and run all the code and cannot support publication as a peer reviewed study in GMD with the current state of data availability.

The ms emulates model runs that are mostly samples of best guesses on choices such as model parameters and structures. This sample is likely missing important uncertainties. Communicating the range of the projections and the discussing "probabilistic uncertainty analyses" (abstract) can lead to unfortunate misunderstandings by the users. This needs to be discussed and the communication made more clear to help the users of this tool and the results to better understand the caveats. There is a nice start towards this goal on the bottom of page 9, but this needs to be made much more clear to the reader (e.g., in the abstract). This potential misunderstanding is the even strengthened in the design of the projections, that uses a "probabilistic MAGICC design" (p. 14). For example, Figure A1 discusses a 66% range. One question the ms should address is how the ranges they present compare to expert assessments (that are cited in the ms).

The ms makes several broad and somewhat ambiguous claims that need to be updated. For example, the ms states: "This MAGICC sea level model has been designed to emulate the behavior of the latest available process-based sea level projections" (p2). This does seem to be in contrast with the findings of DeConto and Pollard (2016), which the ms cites but in my reading not considers. This also needs to flow into the discussion of ice sheet dynamics on page 13 as well as the summary in the discussion (p. 15).

How are "semi-empirical models" (p2) specifically defined? Has there been no use before 2007 (as the ms seems to imply)?

Emulators and semi-empirical models are not necessarily alternatives, as the ms seems to imply (p.2). Overall, the ms does not do justice to the very large body of literature on the design, use, and potential usefulness of emulators in the field of climate change research. This comes back to haunt the ms in the discussion (e.g., page 16). Please add to the introduction a more careful review on this issue and revise the statements on this issue in the discussion. How is "robust" (p.2) defined?

"In this study, we provide a first series of updates for MAGICC version 7 which will be consistent with the ensemble output of CMIP5 " (p. 3). Is this a statement of fact or about a possible future? "The tuned model captures key features of the individual

ocean heat uptake and vertical redistribution behavior of every CMIP5 model " (p. 4). Which ones?

Please add a table of all model parameters, definitions, units, and values.

Please review the formatting of all equations (e.g., equation 3) and symbol uses in the text (e.g., p6 l13).

Why select only 9 parameters for the calibration? How are the chosen? What would happen if you use all parameters? (p. 11)

Why do you choose the 5000 random parameters (I assume as a starting point)? (p. 11). Is this because there are concerns about / evidence for multiple maxima? You should (i) cite and discuss the ground-breaking paper: Hargreaves, J. C., and J. D. Annan (2002), Assimilation of paleo-data in a simple Earth system model, Climate Dynamics, 19(5-6), 371-381 and (ii) provide evidence for the assessment that the chosen method has a decent chance to be close to the global maximum. How are the weights for the RSS chosen (p.11, I also assume Table 1)? Is this choice consistent with the properties of the residuals?

The claim that the presented results are "superior" (p. 16) is unclear and not backed up by evidence. Please define specific metrics (maybe hindcast cross validation error, etc.) and provide the evidence for this.

Please review the format and missing information in the citations.

Please show the residuals for the calibration (Figure 2). Is there a discrepancy in the lower layers? If so, how is this handled in the calibration?

The fonts and line sizes in Figure 3 are too small to read. Please provide readable font sizes and line / symbol separations (as SOM if need be). Why do you choose only the 90% range for Antarctic solid ice discharge? What explains the change in range in several of the panels (e.g., a and c)? Please show the hindcasts and the residuals in for the results shown in Figure 4.

---

## Author Comment (AC1) · 24 Mar 2017

Thank you for pointing us to the necessary title revision. Previous MAGICC model versions provided sea level estimates based on varying sea level components (Wigley and Raper 1987, Wigley and Raper 1992, Wigley 1995, Wigley and Raper 2005). These older developments can be seen as MAGICC sea level model versions 1.0, 1.1, 1.2, 1.3. Here, we here present a new, fully revised MAGICC sea level model which will be also part of the coming MAGICC model version release. Thus, we suggest the following title: "*Synthesizing long-term sea level rise projections – the MAGICC sea level model v2.0*"

References:

Wigley, T. M. L. S. C. B. Raper (1987) Thermal expansion of sea water associated with

global warming. Nature, 330, 127-131.

Wigley, T. M. L.  S. C. B. Raper (1992) Implications for climate and sea level of revised IPCC emissions scenarios. Nature, 357, 293-300.

Wigley, T. M. L. (1995) Global-mean temperature and sea level consequences of green-house gas concentration stabilization. Geophysical Research Letters, 22, 45-48.

Wigley, T. M. L.  S. C. B. Raper (2005) Extended scenarios for glacier melt due to anthropogenic forcing. Geophysical Research Letters, 32.
* * *

---

## Author Comment (AC2) · 24 Mar 2017

**Referee 1 General Comment (RC1.00):** In this paper, the authors describe the first version of the MAGICC sea level model, a set of sea-level emulators linked to versions 6 and 7 of the MAGICC climate model. They provide a clear description of the algorithms for the different sea-level components and of the model calibration. Accordingly, I believe the paper is worth publishing and have only minor comments regarding the paper itself. I suggest the authors add some discussion placing the MAGICC sea level model in the context of other similar tools, such as the BRICK model, also currently in review at GMD (http://dx.doi.org/10.5194/gmd-2016-303). A comparison of the model results to that of other simple sea-level models (e.g., Kopp et al., 2016, and Mengel et al., 2016) under similar forcing would be helpful – prima facie, the projections for 2100 seem to align well with these simpler models.

**Author Response (AR1.00):** We would like to thank the referee for reviewing this manuscript and are encouraged by the positive feedback and the overall recommendation to publish the work with minor revisions. We have extended the discussion on alternative sea level modeling approaches and included hindcast information with comparison datasets, as also requested by referee 2 (Figure A.2). Unfortunately, we find it hard to compare our results directly to the BRICK model, because we could not find any suitable sea level projection time series in the BRICK discussion paper. However, we very much support the efforts by the author team to provide a simple and transparent sea level model framework and have now referenced the discussion paper in our introduction (p.3 l.5). It is encouraging that also other simple model frameworks are in line with recent comprehensive reference assessments. Our key objective was to develop a simple but comprehensive sea level model that emulates the process-based reference data from the Fifth Assessment Report of the IPCC. Consequently, mainly IPCC datasets are used for comparison in the manuscript. Please note that we discovered a bug in the MAGICC routine to prescribe annual mean surface air temperature as well as a problem with the Greenland SID functional form, while carrying out additional checks of the calibration routines after the submission of the manuscript. Also, one specific CMIP5 model input had to be removed due to quality issues with the reference data (BCC-CSM1.1M). All issues have been resolved. The revised prescribed temperature routine required the re-calibration of all sea level model components, with all figures now showing the revised sea level model results. The updated optimal parameter sets are provided in the corresponding Tables. For the Greenland SID parameterization, the calibrated constant was removed, because it prohibited the reproduction of lower bound projections (see revised Equation 6). In addition, the routine was adapted to allow for hindcasts without depleting the maximum outlet glacier volume, which was determined based on the year 2000. The code repository has been updated to account for these changes. All revised and additional Figures as well as one additional Appendix Table are provided at the end of this document.

**RC1.01:** Line 28: I believe the citation here should be to Levermann et al. (2014), not Levermann et al. (2013).

**AR1.01:** Well spotted, thank you! The citation is updated.

**RC1.02:** Line 30: Note that an early semiempirical model was introduced by Gornitz et al. (1982) twenty five years earlier. Gornitz, V., S. Lebedeff, and J. Hansen, 1982: Global sea level trend in the past century. Science, 215, 1611–1614, doi:10.1126/science.215.4540.1611.

**AR1.02:** Thank you for this correct remark. We have revised the text accordingly. The passage on p.2 l.30 now reads: "*In the 1980s, first Semi-Empirical Models (SEMs), which estimate global sea level changes based on the evolution of global mean temperature, were introduced together with early attempts to model thermal expansion based on simplified ocean processes (Gornitz et al. 1982). Generally, SEMs establish statistical relationships between observed/reconstructed global mean temperature or radiative forcing changes and observed/reconstructed global mean sea level changes. Assuming that such relationships stay the same for the future, they are used to estimate future SLR from projected global temperature/forcing changes (Jevrejeva, Moore and Grinsted 2010, Kopp et al. 2016, Rahmstorf 2007, Vermeer and Rahmstorf 2009). As such, these SEMs do not calculate sea level by resolving the underlying physical processes.*"

**RC1.03:** I also note that the code for the sea level model, while available at the gitlab link provided, does not run without the MAGICC model, for which code is not available. I further note that GMD policy states: "If the authors cannot or do not wish to make the code and/or data public (e.g. copyright or licensing restrictions), the reasons must be clearly stated. Note that, for the purpose of the review, the code and/or data must still

be made available to the editor. Access must also be granted to the reviewers whilst preserving their anonymity, if this is legally possible." I am not sure whether the current code availability – dependent upon code that was not available for the review process, and for whose non-availability no explanation was provided – satisfies this policy.

**AR1.03:** We acknowledge that the parent MAGICC model is needed to fully reproduce results with the MAGICC sea level model and regret not providing the source code and a test example with the first version of this manuscript. In order to facilitate the review of the sea level model implementation and to fully support the GMD policies for reproducibility, we have now provided a minimum MAGICC model setup including source code to the editor for distribution to the referees. The sea level model presented in this manuscript is available as open source code on a dedicated gitlab repository, in accordance with the GMD code and data policies. Please see Meinshausen, Raper and Wigley (2011a) and Meinshausen, Wigley and Raper (2011b) for a detailed description of the parent model. MAGICC itself has not been designed as an open-source model, as its initial development and application go back three decades to a time when the open source concept wasn't well established in the scientific community. The source code for MAGICC model version 6 is available under a license agreement on request. Unfortunately, we did not make this clear in the documentation on the digital repository. This has been corrected. We hope that these efforts address the concerns voiced by the referees and allow us to provide a first MAGICC model component on an open-source platform.

References:

Gornitz, V., S. Lebedeff & J. Hansen (1982) Global Sea Level Trend in the Past Century. Science, 215, 1611.

Jevrejeva, S., J. C. Moore & A. Grinsted (2010) How will sea level respond to changes in natural and anthropogenic forcings by 2100? Geophysical Research Letters, 37,

L07703.

Kopp, R. E., A. C. Kemp, K. Bittermann, B. P. Horton, J. P. Donnelly, W. R. Gehrels, C. C. Hay, J. X. Mitrovica, E. D. Morrow & S. Rahmstorf (2016) Temperature-driven global sea-level variability in the Common Era. Proceedings of the National Academy of Sciences, 113, E1434-E1441.

Levermann, A., R. Winkelmann, S. Nowicki, J. L. Fastook, K. Frieler, R. Greve, H. H. Hellmer, M. A. Martin, M. Meinshausen, M. Mengel, A. J. Payne, D. Pollard, T. Sato, R. Timmermann, W. L. Wang & R. A. Bindschadler (2014) Projecting Antarctic ice discharge using response functions from SeaRISE ice-sheet models. Earth Syst. Dynam., 5, 271-293.

Meinshausen, M., S. C. B. Raper & T. M. L. Wigley (2011a) Emulating coupled atmosphere-ocean and carbon cycle models with a simpler model, MAGICC6-Part 1: Model description and calibration. Atmospheric Chemistry and Physics, 11, 1417-1456.

Meinshausen, M., T. M. L. Wigley & S. C. B. Raper (2011b) Emulating atmosphere-ocean and carbon cycle models with a simpler model, MAGICC6-Part 2: Applications. Atmospheric Chemistry and Physics, 11, 1457-1471.

Rahmstorf, S. (2007) A semi-empirical approach to projecting future sea-level rise. Science, 315, 368-370

Vermeer, M. & S. Rahmstorf (2009) Global sea level linked to global temperature. Proceedings of the National Academy of Sciences of the United States of America, 106, 21527-21532.

Please also note the supplement to this comment:
http://www.geosci-model-dev-discuss.net/gmd-2016-233/gmd-2016-233-AC2-supplement.pdf

[Figure]

[Figure]

**Fig. 1.** Figure 2. Potential ocean temperature depth profiles for MAGICC and reference CMIP5 warming under RCP2.6, RCP4.5, RCP6.0 and RCP8.5 scenarios, 2081-2100 anomalies with respect to 1986-2005. [...]

[Figure]

**Fig. 2.** Figure 3. MAGICC sea level model calibration results for thermal expansion (a-d), global glaciers (e-h), Greenland surface mass balance (j-k) and solid ice discharge (l-m) [...]

[Figure]

**Fig. 3.** Figure A.1. Potential ocean temperature residuals for calibrated MAGICC and reference CMIP5 ocean warming under RCP2.6, RCP4.5, RCP6.0 and RCP8.5 scenarios. [...]

[Figure]

**Fig. 4.** Figure A.2. Annual RCP4.5 ocean warming anomalies for the CMIP5 models GISS-E2-R (1850-2300), HadGEM2-CC (1850-2100), and CCSM (1850-2300), relative to 1850 and globally averaged. [...]

[Figure]

**Fig. 5.** Figure A.3. Historical modelled and observed SLR from 1900 to 2000, relative to the 1986-2005 mean. Median and 90% uncertainty ranges are shown for the MAGICC hindcast (orange) [...]

---

## Author Comment (AC3) · 24 Mar 2017

**Referee 2 General Comment (RC2.00):** The ms tackles an important and nontrivial point and makes several new contributions. The authors are to be applauded for the efforts and providing the new insights. Alas, the current ms suffers from several (in my view severe, but pretty easy to fix) shortcomings that need to be mitigated to enable a careful review and to hopefully (and eventually) rise to the level of quality one expects from papers in GMD. These specific points are discussed below. The ms requires in my view a careful and responsive revision and re-review. I would be happy to look at this ms again.

**Author Response (AR2.00):** We thank the referee for this thorough review leading to numerous revisions and additions, which, in our eyes, have substantially improved

the manuscript. We regret not having spent more time on the topic of code availability before the initial submission. We hope that our response and the provision of the MAGICC source code address the issues raised by the referee. Please note, that all revised and additional Figures as well as one additional Appendix Table are provided at the end of this document for your convenience.

**Referee 2 Comment 1(RC2.01):** The ms does, in my view, not follow the data and code policy of GMD. As a reminder, here is the link describing the policy: GMD code and data policy. These rules state: "Preferably, this section should contain the instructions for obtaining the model code and/or data, either from the supplement or from an archive with a digital object identifier (DOI). Suitable repositories can be found at the Registry of Research Data Repositories, e.g. ZENODO for model code. After the paper is accepted, a link to the GMD paper should be added to the metadata of the archive." What is available, as far as I can tell, is code for the sea-level rise model / module. However, it will be extremely tough to reproduce the results in the paper, due to several issues. For example, the interface to call this module is not well documented, nor are the data used to drive and calibrate the model provided. The web-page the ms links to states: "The Fortran code of the model as well as required additional MAGICC input files can be downloaded as a zip-file at the top right of this page. Please note that the MAGICC sea level model is not a stand-alone tool. It is fully integrated into the simple carbon cycle climate model MAGICC and cannot be run separately. The MAGICC model is described in Meinshausen et al. (2011). For a compiled MAGICC version including the sea level model, please contact alexander.nauelsclimate-enegy-college.org. MAGICC is licensed under a Creative Commons Attribution-NonCommercial-ShareAlike3.0 Unported License". Does this mean that the source code for the model needed to run the sea-level model is not accessible? Why is only a compiled version distributed? Given this situation, I would assign a very low score for reproducibility (see the discussion by Haas in the recent Risk Analysis volume (Risk Analysis, Vol. 36, No. 10, 2016, DOI: 10.1111/risa.12730). Many other studies have abided by the nice standards outlined

in GMD and a publication in GMD should clearly abide by these standards. This is a decision for the editor. I cannot carefully review the ms without the ability to see and run all the code and cannot support publication as a peer reviewed study in GMD with the current state of data availability.

**AR2.01:** We acknowledge the concerns expressed by the referee and regret that we have not clarified the availability of the MAGICC source code earlier. We hope that the following update resolves this issue. In line with the GMD code and data policies, the sea level model presented here is available as open-source code on a transparent and version-controlled repository. Once the manuscript is accepted in its final version, we will also deposit the sea level code on ZENODO with a DOI. We agree that it is challenging to reproduce the sea level model output without the parent model code. The MAGICC model itself is not open-source, which is mainly due to the long model development history. The model was first introduced in the 1980s, a time when the open-source concept wasn't well established in the science community. Subsequent MAGICC developments have all been carried out without the scope and resources to fully migrate the model to an open-source platform. Unfortunately, the authors of this manuscript are not in a position to go open source with the code in its current version. The model source-code is available for interested researchers under a non-commercial license agreement on request, which is now communicated correctly in the repository documentation. A MAGICC model package with the source-code including the sea level model has been provided to the editor for distribution to the referees to ensure transparency and reproducibility of the presented work. The most recent MAGICC model version 6 has been extensively documented in Meinshausen et al. (2011a) and Meinshausen et al. (2011b). Data for the atmospheric and ocean variables used for the model calibration can be downloaded from the CMIP5 archive. The interpolated calibration data for the MAGICC ocean model are available on request. This is now clarified in the code and data section (p.17 l.20). The other sea level components are calibrated with additional published datasets that were received by courtesy of the respective author teams. With the sea level model code openly provided and the full

MAGICC model available to the referees as part of this review and for interested peers upon request, we hope that we have addressed the referee's concerns regarding the data and code policy of GMD.

**RC2.02:** The ms emulates model runs that are mostly samples of best guesses on choices such as model parameters and structures. This sample is likely missing important uncertainties. Communicating the range of the projections and the discussing "probabilistic uncertainty analyses" (abstract) can lead to unfortunate misunderstandings by the users. This needs to be discussed and the communication made more clear to help the users of this tool and the results to better understand the caveats. There is a nice start towards this goal on the bottom of page 9, but this needs to be made much more clear to the reader (e.g., in the abstract). This potential misunderstanding is the even strengthened in the design of the projections, that uses a "probabilistic MAGICC design" (p. 14). For example, Figure A1 discusses a 66% range. One question the ms should address is how the ranges they present compare to expert assessments (that are cited in the ms).

**AR2.02:** The referee is correct in noting that the presented sea level model does not account for deep uncertainties, i.e., uncertainties due to processes that are not yet known or not captured by process-based models. It is not probabilistic in that sense. Rather, we present an approach to reflect key uncertainties for all major SLR drivers based on the process-based projections presented in the Fifth Assessment Report of the IPCC. Our model is probabilistic in the sense that we combine parameter uncertainties in a way that results in each calibrated parameter set reasonably reproducing the specific reference data. By applying the well-established probabilistic Metropolis-Hastings Markov chain Monte Carlo projection method from Meinshausen et al. (2009), which also underlies, e.g., Meinshausen et al. (2011c), Rogelj, Meinshausen and Knutti (2012), Perrette et al. (2013), Levermann et al. (2014), we can additionally map key climate and carbon-cycle related uncertainties;

above all, the range of equilibrium climate sensitivities presented in IPCC's AR5. The MAGICC sea level emulator therefore provides projections that, besides scenario related uncertainties, reflect model uncertainties included in the reference data (i.e. from the CMIP5 ensemble), as well as uncertainties from carbon-cycle responses (Meinshausen et al. 2011a, Friedlingstein et al. 2014) and latest climate sensitivity estimates (Flato et al. 2013). In order to avoid confusion regarding the scope of uncertainties that can be assessed with the presented approach, we have modified the abstract, with the relevant sentence on p.1 l.18 now reading: "*The MAGICC sea level model provides a flexible and efficient platform for the analysis of major scenario, model, and climate uncertainties underlying long-term SLR projections*". In addition, we have elaborated on the probabilistic method and the comparison of our Antarctic solid ice discharge projections (90% model ranges now included in the Appendix figures) to the ones presented in DeConto and Pollard (2016). We have revised p.14 l.17 to read: "*In order to also provide a sufficiently large number of model runs for 2300, we use 600 historically-constrained parameter sets that have been derived using a probabilistic Metropolis-Hastings Markov chain Monte Carlo method (Meinshausen et al. 2009). This approach has been extended to also reflect carbon-cycle uncertainties (Friedlingstein et al. 2014) and the climate sensitivity range of the latest IPCC assessment (Flato et al. 2013, Rogelj et al. 2012, Rogelj et al. 2014)*"; the section on p.17 l.1 has been updated: "*The Antarctic ice sheet response, for example, could be subject to more rapid, non-linear dynamics that are not captured by the emulated process-based projections. Only recently, DeConto and Pollard (2016) have revised potential future Antarctic contributions to global sea level based on indicators from paleoclimatic archives. For RCP8.5, they project 2100 contributions of around 1m from Antarctica alone, with 2300 contributions reaching up to around 10 m. The MAGICC sea level model projections for the Antarctic SID contribution are based on Levermann et al. (2014) and only yield up to around 0.35 m in 2100 and 2.68 m in 2300 for the upper bound of the 90% range. As the more recent research suggests, these estimates may be too low, indicating that the Antarctic contribution to*

*future SLR is subject to additional uncertainties. This illustrates the need to handle long-term SLR projections with care and to note the corresponding methodological caveats; in particular, those surrounding the representation of Antarctic ice-sheet changes.*"

**RC2.03:** The ms makes several broad and somewhat ambiguous claims that need to be updated. For example, the ms states: "This MAGICC sea level model has been designed to emulate the behavior of the latest available process-based sea level projections" (p2). This does seem to be in contrast with the findings of DeConto and Pollard (2016), which the ms cites but in my reading not considers. This also needs to flow into the discussion of ice sheet dynamics on page 13 as well as the summary in the discussion (p. 15).

**AR2.03:** We agree with the referee that our statement regarding the selected reference datasets was deceptive. We have revised the corresponding statement on p.2 l.13 to read: "*This MAGICC sea level model has been designed to emulate the behavior of process-based sea level projections presented in the IPCC's Fifth Assessment Report (Church et al. 2013a), with thorough calibrations for each major sea level component.*" The discussion now includes a direct comparison of the MAGICC sea level model results with the findings from DeConto and Pollard (2016). After searching for other instances of using the potentially ambiguous adjective "latest", we have also decided to change the land water related statement in the abstract on p.1 l.15 to read: "*The land water storage component replicates recent hydrological modeling results.*"

**RC2.04:** How are "semi-empirical models" (p2) specifically defined? Has there been no use before 2007 (as the ms seems to imply)?

**AR2.04:** We now provide more specific details on the design of semi-empirical models and clarify that the semi-empirical concept has been applied well before 2007. The

section on p.2 l.30 now reads: "*In the 1980s, first Semi-Empirical Models (SEMs), which estimate global sea level changes based on the evolution of global mean temperature, were introduced together with early attempts to model thermal expansion based on simplified ocean processes (Gornitz et al. 1982). Generally, SEMs establish statistical relationships between observed/reconstructed global mean temperature or radiative forcing changes and observed/reconstructed global mean sea level changes. Assuming that such relationships stay the same for the future, they are used to estimate future SLR from projected global temperature/forcing changes (Jevrejeva et al. 2010, Kopp et al. 2016, Rahmstorf 2007, Vermeer and Rahmstorf 2009). As such, these SEMs do not calculate sea level by resolving the underlying physical processes.*"

**RC2.05:** Emulators and semi-empirical models are not necessarily alternatives, as the ms seems to imply (p.2). Overall, the ms does not do justice to the very large body of literature on the design, use, and potential usefulness of emulators in the field of climate change research. This comes back to haunt the ms in the discussion (e.g., page 16). Please add to the introduction a more careful review on this issue and revise the statements on this issue in the discussion. How is "robust" (p.2) defined?

**AR2.05:** In order to define where the presented MAGICC sea level sits methodologically, we have referenced two different methods to efficiently estimate future sea level rise. This distinction is not meant to suggest that semi-empirical models and sea level emulators are alternatives. Both methods have their individual strengths and weaknesses and are often closely related with regards to underlying parameterizations. We have revised the manuscript to include more information on expert elicitations as well as sea level modeling methods that use paleoclimatic information to provide future projections. However, we think that providing a comprehensive discussion on "emulators in the field of climate change research" would unnecessarily lengthen the paper and distract readers interested in the sea level model description. We hope that the referee understands our reluctance to extend the introduction beyond

the revised content. On p.3 l.13, we used "robust" to indicate that the calibrations for all parameterizations converge to reach their global optima (see AR2.10 and AR2.11). We acknowledge that this term is misleading and refined the corresponding statement. The following changes have been made to the text: p.3 l.5: "*Sea level rise projections have also been provided based on expert elicitations (Horton et al. 2014). Furthermore, sea level expert judgments have been combined with statistical models synthesizing sea level projections for individual components (Kopp et al. 2014). Other studies have used an extended suite of methods, analysing paleoclimatic archives, modeling parts of the SLR response with a reduced complexity model, and deriving future projections for land ice contributions based semi-empirical considerations (Clark et al. 2016). The growing efforts in the sea level modeling community to provide fully transparent and freely available model code are reflected by the recent introduction of a transparent, simple model framework to estimate regional sea levels (Wong et al. 2017). Previous MAGICC versions provided sea level rise estimates based on parameterizations for selected components (Wigley 1995, Wigley and Raper 2005, Wigley and Raper 1987, Wigley and Raper 1992). Here, we adopt the approach of deriving a total sea level response by emulating existing process-based projections for individual sea level components (Perrette et al. 2013, Schleussner et al. 2016). Future sea level dynamics are synthesized by calibrating parameterizations to the selected complex model projections for all major sea level contributions. Progress in the understanding of individual sea level processes and the availability of revised future sea level contributions require sea level emulators to be updated regularly. With this study, we are able to complement the existing sea level projection emulators with a platform based on a comprehensive set of individual sea level components that allows for projections consistent with IPCC AR5 estimates.*", p.3 l.13: "*It mimics process-based sea level responses for the seven main sea level components with thoroughly calibrated parameterizations that extend global sea level projections to 2300.*", p.16 l.25: "*Sea level emulators complement the comprehensive but com-putationally expensive, process-based sea level models due to their flexible and*

*efficient design. They can be quickly adapted to, e.g., incorporate previously unknown uncertainties from newly quantified ice sheet processes (Clark et al. 2016, DeConto and Pollard 2016). Being directly coupled to MAGICC, our sea level model can also account for additional climate system response uncertainties and provide consistent projections for a wide range of climate change scenarios beyond the standard IPCC pathways. The latter aspects describe key strengths of the MAGICC sea level model and make it a useful tool to assess SLR for scenarios that are not covered by larger, more comprehensive models. The emulated MAGICC sea level projections reflect, independently, the reference responses for each individual sea level component, assuming that the implemented parameterizations fully capture the process-based simulations. Underlying model uncertainties differ substantially for the individual sea level components (Church et al. 2013b).*"

**RC2.06:** "In this study, we provide a first series of updates for MAGICC version 7 which will be consistent with the ensemble output of CMIP5" (p. 3). Is this a statement of fact or about a possible future?

**AR2.06:** This is a statement of fact. Currently, the MAGICC carbon cycle is being updated. We expect the fully CMIP5-consistent MAGICC version 7 to be published by the end of this year.

**RC2.07:** "The tuned model captures key features of the individual ocean heat uptake and vertical redistribution behavior of every CMIP5 model" (p. 4). Which ones?

**AR2.07:** We use ocean-layer specific potential ocean temperature change as proxy for ocean heat uptake behaviour. As can be seen from Figures 2 and 3, the calibrated MAGICC ocean provides potential ocean temperature warming profiles and thermal expansion projections that are in line with the CMIP5 reference data. We have revised the statement on p.4 l.18: "*The tuned model captures ocean-layer specific thetao*

*change and related vertical redistribution characteristics of individual CMIP5 models, both indicators for overall ocean heat uptake behaviour.*"

**RC2.08:** Please add a table of all model parameters, definitions, units, and values. Please review the formatting of all equations (e.g., equation 3) and symbol uses in the text (e.g., p6 l13).

**AR2.08:** We have now included an overview Table A.1 in the Appendix, listing key calibration variables and the calibration parameters for every sea level component. The calibration values have all been provided before in Tables 1 to 5. Thanks for notifying us about the equation formatting issues. All equations have been checked, equation 2, 3, and 6 have been updated (see also AR2.09). We have included an additional sentence on p.11 l.32 to refer to the new Table: "*For an overview of all relevant variables and calibration parameters please see Table A.1.*"

**RC2.09:** Why select only 9 parameters for the calibration? How are the chosen? What would happen if you use all parameters? (p. 11)

**AR2.09:** For the MAGICC ocean model calibration, we have selected all MAGICC parameters that directly determine the ocean-layer specific potential ocean temperature and corresponding thermal expansion responses. These 9 parameters drive the band-routine of the hemispheric upwelling-diffusion ocean model and are now also listed in the additional Appendix Table A.1 for clarity (see AR2.08). The MAGICC sea level model calibration has been carried out per sea level contribution, only optimising the free parameters of the respective component, while prescribing the annual mean surface temperatures. As such, it was easier for us to ensure consistency with the reference data, separate the effects of individual parameter variations, and analyse the calibration quality. While carrying out additional checks of the calibration routines after the submission of the manuscript, we unfortunately discovered a bug in the

MAGICC routine to prescribe annual mean surface air temperature as well as a problem with the Greenland SID functional form. Also, one specific CMIP5 model input had to be removed due to quality issues with the reference data (BCC-CSM1.1M). All issues have been resolved. The revised prescribed temperature routine required the re-calibration of all sea level model components, with all figures now showing the corrected sea level model results. The updated optimal parameter sets are provided in the corresponding Tables. For the Greenland SID parameterisation, the calibrated constant was removed, because it prohibited reproducing lower bound projections (see revised Equation 6). In addition, the routine was adapted to allow for hindcasts without depleting the maximum outlet glacier volume, which was determined based on the year 2000. The code repository has been updated to account for these changes. We have revised the following sections to clarify the ocean model parameter selection and to account for the necessary code revisions: p.11 l.11 "*Here, we select all MAGICC parameters that directly influence the ocean-layer specific potential ocean temperature and corresponding thermal expansion responses. These 9 parameters drive the band-routine of the hemispheric upwelling-diffusion ocean model*", p.7 l.25 "*Applying the scaling suggested by Church et al. (2013a), our total Greenland SID maximum ice discharge volumes amount to around 180 mm and 268 mm SLE for the minimum and maximum cases presented in Nick et al. (2013)*", p.15 l.11 "*The overall historically-constrained, probabilistic MAGICC global mean tas response for the 21st century is stronger than in the CMIP5 reference data for RCP4.5, RCP6.0 and RCP8.5 scenarios. This slightly steeper 21st century global mean tas slope is also reflected in the corresponding probabilistic MAGICC 2100 SLR estimates, given the strong air temperature dependence of the sea level model (see panels (a) to (d) of Figure 4).*"

**RC2.10:** Why do you choose the 5000 random parameters (I assume as a starting point)? (p. 11). Is this because there are concerns about / evidence for multiple maxima?

**AR2.10:** Yes, there were concerns about local optima or too flat gradients for the optimization routine to converge properly. Initial calibration testing with 100 initial random runs pointed to local convergence of the MAGICC ocean parameter optimization routine. Extensive testing with 500, 1000, 5000, 7500, and 10000 initial random runs was conducted to establish the necessary number of pre-optimization random parameter sets to appropriately map the parameter space. While all tests with 1000 parameters showed global convergence, we opted for 5000 random runs to ensure a sufficiently large set of samples for deriving the initial parameter set of the optimization routine. The text on p.11 l.22 has been revised: "*5000 random parameter sets are drawn prior to each model optimization procedure. The number of initial random runs has been determined through iterative testing to ensure convergence to a global optimum. The resulting best fit is subsequently used for the initialization of the automated Nelder-Mead simplex optimization routine (Lagarias et al. 1998, Nelder and Mead 1965) with a termination tolerance of $10\text{-}8$ and a maximum iteration number of 10,000.*"

**RC2.11:** You should (i) cite and discuss the ground-breaking paper: Hargreaves, J. C., and J. D. Annan (2002), Assimilation of paleo-data in a simple Earth system model, Climate Dynamics, 19(5-6), 371-381 and (ii) provide evidence for the assessment that the chosen method has a decent chance to be close to the global maximum. How are the weights for the RSS chosen (p.11, I also assume Table 1)? Is this choice consistent with the properties of the residuals?

**AR2.11:** Thank you for pointing us to this relevant paper which is now cited in the revised draft. The Nelder-Mead Simplex optimisation routine as described in Lagarias et al. (1998) has been successfully applied for previous MAGICC calibration procedures (Meinshausen et al. 2011a). We are using 5000 iterations of randomly drawn calibration parameter sets to ensure that the subsequent optimisation routine shows global convergence in providing the optimal parameter fits. It is only for the

ocean calibration that weights different from 1 have been applied. The different units of the ocean target variables (K for potential ocean temperatures per layer, mm for thermal expansion) call for a sensible choice of variable specific weights to not introduce an optimisation error bias for the RSS. To ensure that the calibration routine prioritises the ocean calibration while still globally converging for the thermal expansion scaling coefficient, we have iteratively derived the thermal expansion weight (0.001) and the ocean layer temperature anomaly weight (10). The following sections of the calibration section have been revised: p.11 l.11: "*Previous studies have shown that highly parameterised simple models do successfully show global convergence when calibrating a large number of free parameters (Hargreaves and Annan 2002, Meinshausen et al. 2011a)*", p.11 l.26: "*The zostoga optimization component is given four orders of magnitude less relative weight than the thetao component in order to prioritize the accurate layer-by-layer emulation of the respective CMIP5 model thetao time series.*"

**RC2.12:** The claim that the presented results are "superior" (p. 16) is unclear and not backed up by evidence. Please define specific metrics (maybe hindcast cross validation error, etc.) and provide the evidence for this.

**AR2.12:** We admit that "superior" should not be used in this context. However, we still think that the MAGICC sea level model has advantages over other simple sea level models that either provide a global sea level response only, e.g. semi-empirical approaches like Rahmstorf (2007), use less up-to-date reference data (Perrette et al. 2013, Schleussner et al. 2016), or do not reproduce historical data as well (Mengel et al. 2016). We have revised the corresponding statement on p.16 l.3 to now read: "*The close reproduction of selected reference data (Figure 3, Figure A.3), the consistent translation of climate forcing into a SLR response within the MAGICC model, and the comprehensive representation of relevant processes (e.g. the thermal expansion contribution produced by the CMIP5-consistent MAGICC ocean model and*

*the inclusion of the land water storage sea level component) make the MAGICC sea
level model a powerful addition to the existing sea level emulators.*"

**RC2.13:** Please review the format and missing information in the citations.

**AR2.13:** Done. We have updated references which were either incomplete or not in
the right format.

**RC2.14:** Please show the residuals for the calibration (Figure 2). Is there a discrepancy
in the lower layers? If so, how is this handled in the calibration?

**AR2.14:** The calibration residuals for Figure 2 are now provided as potential ocean
temperature anomalies for every MAGICC ocean layer in the Appendix, Figure A.1.
In addition we have decided to highlight specific model outliers in Figure 2 if they
are not covered by the 90% CMIP5 reference and resulting 90% MAGICC model
ranges. These figures now more clearly show what has already been stated on p.12
l.20, i.e. the tendency of the MAGICC bottom layers to warm more than the CMIP5
reference data. Calibration results for 2 of the 36 CMIP5 models show a major bottom
layer warming bias. The GISS-E2-R reference data shows strong mid-layer warming
combined with actual bottom layer cooling, the HadGEM2-CC data shows cooling in
the upper 500 m over the historical period. In both cases, the MAGICC hemispheric
upwelling-diffusion ocean model cannot fully capture these characteristics and, for the
HadGEM2-CC model, overcompensates the surface cooling with strong bottom layer
warming. The calibration routine itself gives all ocean layers the same weight and
terminates the optimisation when the reduction in the RSS stays with the 10-8 termina-
tion tolerance. Both model calibration results represent the optimal parameter fits for
the given reference data. We have revised the text to more clearly state these aspects.
p.12 l.17 now reads: "*The Figure also provides information on individual model outliers
for reference data and calibration results. Corresponding potential ocean temperature*

*residuals are shown in Figure A.1*", p.12 l.18: "*The updated MAGICC ocean deviates from the CMIP5 data in a few cases. Generally, there appears to be less warming in the mid-ocean between around 1500 m and 2500 m than in the CMIP5 reference data. Also, there is a tendency for the MAGICC bottom layers to warm more than the CMIP5 reference data. However, it is only for 2 of the 36 CMIP5 models used that calibration results show a major bottom layer warming bias. The GISS-E2-R reference data show strong mid-layer warming combined with actual bottom layer cooling, while the HadGEM2-CC data show cooling in the upper 500 m over the historical period (Figure A.2). In both cases, the MAGICC hemispheric upwelling-diffusion ocean model cannot fully capture these characteristics. For the HadGEM2-CC emulation, MAGICC overcompensates the surface cooling with strong bottom layer warming.*" Please see also AR2.09, AR2.10, and AR2.11 for the calibration related part of the referee comment.

**RC2.15:** The fonts and line sizes in Figure 3 are too small to read. Please provide readable font sizes and line / symbol separations (as SOM if need be). Why do you choose only the 90% range for Antarctic solid ice discharge? What explains the change in range in several of the panels (e.g., a and c)? Please show the hindcasts and the residuals for the results shown in Figure 4.

**AR2.15:** Thanks for pointing this out. We have increased font size, line size, and contrast where possible to improve the readability of the Figure. Hopefully, the editor will agree to provide it as a full page graphic. We have purposefully designed the Figure in the current format to provide all calibration results in one place and to allow for immediate visual checks of the calibration quality. We would like to keep this format. The ranges in each panel have been adapted to fit the projection magnitude and reference data time frame for each sea level component. This has now been stated in the caption. We only show the 90% range of the Antarctic SID component because we were not able to receive more detailed comparison data from Levermann

et al. (2014). The hindcast for total SLR is now provided in Figure A.3 with three selected comparison data sets from Mengel et al. (2016), Hay et al. (2015), and Church and White (2011). The author teams of the observational datasets have been included in the acknowledgements. Hindcast for the individual SLR components have been included in Figures A.4 to A.7. For the Greenland SMB, SID and the Antarctic SMB components, the parameterisations are run freely for the historical period as the process-based model projections used as reference data do not cover this period. The historical sea level response, in particular for the GIS SID component, could be further improved in the future by also including reanalyses or observational calibration data as part of a sea level model update. This, however, is beyond the scope of the present work. We updated the text and now refer to the newly provided hindcast on p.15 l.5: "*In Figure A.3, we provide MAGICC SLR hindcast results and three comparison datasets for the period 1900 to 2000. The MAGICC sea level model shows good agreement with the observational datasets based on Church and White (2011) and Hay et al. (2015). The global 1900-2300 SLR responses are provided for all RCPs and each sea level component in the Appendix Figures A.4 to A.7.*" Additional text has been added to the discussion section on p.16 l.3: *Our sea level model transparently emulates and combines long-term sea level projections from process-based models. It also reproduces well observed past total sea level change (see Figure A.3). Therefore, we are confident that this model contributes to advancing efficient long-term sea level projections.*"

References:

Church, J. A., P. U. Clark, A. Cazenave, J. M. Gregory, S. Jevrejeva, A. Levermann, M. A. Merrifield, G. A. Milne, R. S. Nerem, P. D. Nunn, A. J. Payne, W. T. Pfeffer, D. Stammer & A. S. Unnikrishnan. 2013a. Sea Level Change. In Climate Change 2013: The Physical Science Basis. Contribution of Working Group I to the Fifth Assessment Report of the Intergovernmental Panel on Climate Change, ed. T. F. Stocker, D. Qin, G.-K. Plattner, M. Tignor, S.K. Allen, J. Boschung, A. Nauels, Y. Xia, V. Bex and P.M.

Midgley. Cambridge University Press, Cambridge, United Kingdom and New York, NY, USA.

Church, J. A., D. Monselesan, J. M. Gregory & B. Marzeion (2013b) Evaluating the ability of process based models to project sea-level change. Environmental Research Letters, 8, 014051. Church, J. A. N. J. White (2011) Sea-Level Rise from the Late 19th to the Early 21st Century. Surveys in Geophysics, 32, 585-602.

Clark, P. U., J. D. Shakun, S. A. Marcott, A. C. Mix, M. Eby, S. Kulp, A. Levermann, G. A. Milne, P. L. Pfister, B. D. Santer, D. P. Schrag, S. Solomon, T. F. Stocker, B. H. Strauss, A. J. Weaver, R. Winkelmann, D. Archer, E. Bard, A. Goldner, K. Lambeck, R. T. Pierrehumbert & G.-K. Plattner (2016) Consequences of twenty-first-century policy for multi-millennial climate and sea-level change. Nature Climate Change, 6, 360–369.

DeConto, R. M. & D. Pollard (2016) Contribution of Antarctica to past and future sea-level rise. Nature, 531, 591-597.

Flato, G., J. Marotzke, B. Abiodun, P. Braconnot, S. C. Chou, W. Collins, P. Cox, F. Driouech, S. Emori, V. Eyring, C. Forest, P. Gleckler, E. Guilyardi, C. Jakobs, V. Kattsov, C. Reason & M. Rummukainen. (2013) Evaluation of Climate Models. In Climate Change 2013: The Physical Science Basis. Contribution of Working Group I to the Fifth Assessment Report of the Intergovernmental Panel on Climate Change, ed. T. F. Stocker, D. Qin, G.-K. Plattner, M. Tignor, S.K. Allen, J. Boschung, A. Nauels, Y. Xia, V. Bex and P.M. Midgley. Cambridge University Press, Cambridge, United Kingdom and New York, NY, USA.

Friedlingstein, P., M. Meinshausen, V. K. Arora, C. D. Jones, A. Anav, S. K. Liddicoat & R. Knutti (2014) Uncertainties in CMIP5 Climate Projections due to Carbon Cycle Feedbacks. Journal of Climate, 27, 511-526.

Gornitz, V., S. Lebedeff & J. Hansen (1982) Global Sea Level Trend in the Past Century. Science, 215, 1611.

Hargreaves, J. & J. Annan (2002) Assimilation of paleo-data in a simple Earth system model. Climate Dynamics, 19, 371-381.

Hay, C. C., E. Morrow, R. E. Kopp J. X. Mitrovica (2015) Probabilistic reanalysis of twentieth-century sea-level rise. Nature, 517, 481-484.

Horton, B. P., S. Rahmstorf, S. E. Engelhart & A. C. Kemp (2014) Expert assessment of sea-level rise by AD 2100 and AD 2300. Quaternary Science Reviews, 84, 1-6.

Jevrejeva, S., J. C. Moore & A. Grinsted (2010) How will sea level respond to changes in natural and anthropogenic forcings by 2100? Geophysical Research Letters, 37, L07703.

Kopp, R. E., R. M. Horton, C. M. Little, J. X. Mitrovica, M. Oppenheimer, D. J. Rasmussen, B. H. Strauss & C. Tebaldi (2014) Probabilistic 21st and 22nd century sea-level projections at a global network of tide-gauge sites. Earth's Future, 2, 383-406.

Kopp, R. E., A. C. Kemp, K. Bittermann, B. P. Horton, J. P. Donnelly, W. R. Gehrels, C. C. Hay, J. X. Mitrovica, E. D. Morrow & S. Rahmstorf (2016) Temperature-driven global sea-level variability in the Common Era. Proceedings of the National Academy of Sciences, 113, E1434-E1441.

Lagarias, J., J. Reeds, M. Wright & P. Wright (1998) Convergence Properties of the Nelder–Mead Simplex Method in Low Dimensions. SIAM Journal on Optimization, 9, 112-147.

Levermann, A., R. Winkelmann, S. Nowicki, J. L. Fastook, K. Frieler, R. Greve, H. H. Hellmer, M. A. Martin, M. Meinshausen, M. Mengel, A. J. Payne, D. Pollard, T. Sato, R. Timmermann, W. L. Wang & R. A. Bindschadler (2014) Projecting Antarctic ice discharge using response functions from SeaRISE ice-sheet models. Earth Syst. Dynam., 5, 271-293.

Meinshausen, M., N. Meinshausen, W. Hare, S. C. B. Raper, K. Frieler, R. Knutti, D. J. Frame & M. R. Allen (2009) Greenhouse-gas emission targets for limiting global

warming to 2 degrees C. Nature, 458, 1158-1196.

Meinshausen, M., S. C. B. Raper & T. M. L. Wigley (2011a) Emulating coupled atmosphere-ocean and carbon cycle models with a simpler model, MAGICC6-Part 1: Model description and calibration. Atmospheric Chemistry and Physics, 11, 1417-1456.

Meinshausen, M., T. M. L. Wigley & S. C. B. Raper (2011b) Emulating atmosphere-ocean and carbon cycle models with a simpler model, MAGICC6-Part 2: Applications. Atmospheric Chemistry and Physics, 11, 1457-1471.

Meinshausen, M., S. J. Smith, K. Calvin, J. S. Daniel, M. L. T. Kainuma, J. F. Lamarque, K. Matsumoto, S. A. Montzka, S. C. B. Raper, K. Riahi, A. Thomson, G. J. M. Velders & D. P. P. van Vuuren (2011c) The RCP greenhouse gas concentrations and their extensions from 1765 to 2300. Climatic Change, 109, 213-241.

Mengel, M., A. Levermann, K. Frieler, A. Robinson, B. Marzeion & R. Winkelmann (2016) Future sea level rise constrained by observations and long-term commitment. Proceedings of the National Academy of Sciences, 113, 2597-2602.

Nelder, J. A. & R. Mead (1965) A Simplex Method for Function Minimization. The Computer Journal, 7, 308-313.

Nick, F. M., A. Vieli, M. L. Andersen, I. Joughin, A. Payne, T. L. Edwards, F. Pattyn & R. S. W. van de Wal (2013) Future sea-level rise from Greenland/'s main outlet glaciers in a warming climate. Nature, 497, 235-238.

Perrette, M., F. Landerer, R. Riva, K. Frieler & M. Meinshausen (2013) A scaling approach to project regional sea level rise and its uncertainties. Earth System Dynamics, 4, 11-29.

Rahmstorf, S. (2007) A semi-empirical approach to projecting future sea-level rise. Science, 315, 368-370.

[Figure]

Rogelj, J., M. Meinshausen & R. Knutti (2012) Global warming under old and new scenarios using IPCC climate sensitivity range estimates. Nature Clim. Change, 2, 248-253.

Rogelj, J., M. Meinshausen, J. Sedlacek & R. Knutti (2014) Implications of potentially lower climate sensitivity on climate projections and policy. Environmental Research Letters, 9, 031003.

Schleussner, C. F., T. K. Lissner, E. M. Fischer, J. Wohland, M. Perrette, A. Golly, J. Rogelj, K. Childers, J. Schewe, K. Frieler, M. Mengel, W. Hare & M. Schaeffer (2016) Differential climate impacts for policy-relevant limits to global warming: the case of 1.5C and 2C. Earth Syst. Dynam., 7, 327-351.

Vermeer, M. & S. Rahmstorf (2009) Global sea level linked to global temperature. Proceedings of the National Academy of Sciences of the United States of America, 106, 21527-21532.

Wigley, T. M. L. (1995) Global-mean temperature and sea level consequences of greenhouse gas concentration stabilization. Geophysical Research Letters, 22, 45-48.

Wigley, T. M. L. & S. C. B. Raper (1987) Thermal expansion of sea water associated with global warming. Nature, 330, 127-131.

Wigley, T. M. L. & S. C. B. Raper (1992) Implications for climate and sea level of revised IPCC emissions scenarios. Nature, 357, 293-300.

Wigley, T. M. L. & S. C. B. Raper (2005) Extended scenarios for glacier melt due to anthropogenic forcing. Geophysical Research Letters, 32.

Wong, T. E., A. Bakker, K. Ruckert, P. Applegate, A. Slangen & K. Keller (2017) BRICK v0.1, a simple, accessible, and transparent model framework for climate and regional sea-level projections. Geosci. Model Dev. Discuss., 2017, 1-36.

Please also note the supplement to this comment:

http://www.geosci-model-dev-discuss.net/gmd-2016-233/gmd-2016-233-AC3-supplement.pdf

[Figure]

**Fig. 1.** Figure 2. Potential ocean temperature depth profiles for MAGICC and reference CMIP5 warming under RCP2.6, RCP4.5, RCP6.0 and RCP8.5 scenarios, 2081-2100 anomalies with respect to 1986-2005. [...]

**Fig. 2.** Figure 3. MAGICC sea level model calibration results for thermal expansion (a-d), global glaciers (e-h), Greenland surface mass balance (j-k) and solid ice discharge (l-m) [...]

[Figure]

**Fig. 3.** Figure A.1. Potential ocean temperature residuals for calibrated MAGICC and reference CMIP5 ocean warming under RCP2.6, RCP4.5, RCP6.0 and RCP8.5 scenarios. [...]

[Figure]

**Fig. 4.** Figure A.2. Annual RCP4.5 ocean warming anomalies for the CMIP5 models GISS-E2-R (1850-2300), HadGEM2-CC (1850-2100), and CCSM (1850-2300), relative to 1850 and globally averaged. [...]

Figure with legend:
- MAGICC sea level model (orange dashed)
- Mengel et al. 2016 (dotted)
- Hay et al. 2015 (blue)
- updated Church et al. 2011 (black)

Y-axis: sea level rise wrt 1986-2005 [mm]
X-axis: Year

**Fig. 5.** Figure A.3. Historical modelled and observed SLR from 1900 to 2000, relative to the 1986-2005 mean. Median and 90% uncertainty ranges are shown for the MAGICC hindcast (orange) [...]

**Supplement:**

**Table A.1.** List of variables and free parameters used for the individual MAGICC sea level component calibrations.

| Climate variables | Unit | Description |
|---|---|---|
| *tas* | $K$ | surface air temperature |
| *thetao* | $K$ | potential ocean temperature |
| *zostoga* | $mm$ | thermal expansion |
| **MAGICC ocean paramters** | | |
| $K_z$ | $\frac{cm^2}{s}$ | vertical thermal diffusivity |
| $\frac{dK_{z_{top}}}{dT}$ | $\frac{cm^2}{sK}$ | sensitivity to global mean *tas* at the mixed layer boundary |
| $\eta$ | $K$ | sea-ice adjustment offset |
| $\gamma$ | $\frac{1}{K}$ | sea-ice adjustment factor |
| $w_0$ | $\frac{m}{yr}$ | initial upwelling velocity |
| $\beta$ | | ratio of changes in temperature of entraining waters to polar sinking waters |
| $\frac{\Delta w_t}{w_t}$ | | ratio of variable to fixed upwelling for every time step |
| $T_{w_t}$ | $K$ | threshold temperatures for constant upwelling rates |
| $\phi$ | | global thermal expansion scaling |
| **Glacier parameters** | | |
| $\kappa$ | $\frac{1}{K}$ | glacier sensitivity |
| $\nu$ | | temperature sensitivity exponent |
| **Greenland SMB parameters** | | |
| $\upsilon$ | $\frac{mm}{K}$ | temperature sensitivity |
| $\chi$ | | relative magnitude of linear and non-linear terms |
| $\varphi$ | | temperature sensitivity exponent |
| **Greenland SID parameters** | | |
| $\varrho$ | | discharge sensitivity |
| $\epsilon$ | $\frac{1}{K}$ | temperature sensitivity |
| $GIS_{max}^{outlet}$ | $mm$ | maximum Greenland outlet glacier volume |
| **Antarctic SMB parameters** | | |
| $\xi$ | $\frac{mm}{K}$ | temperature sensitivity |
| $\rho$ | | relative magnitude of linear and non-linear terms |
| $\sigma$ | | temperature sensitivity exponent |

---

## Author Response (AR2)

**GMD-2016-233 Author Responses to Editor Review (Minor Revisions)**

Dear David,

We are delighted that our manuscript will be published by GMD subject to minor revisions. We acknowledge the challenges that arise from not being able to provide the full MAGICC model as open source. However, we hope that the publication of the sea level model source code and extended MAGICC input data as well as model configuration information on two Zenodo repositories will further improve the reproducibility of our work.

Please find our responses together with the corresponding manuscript changes below. The second part of the document includes the revised manuscript with all corresponding changes marked in yellow.

In addition to the editor comments, we have addressed the additional comments by referee #2. Also, a few rounding inaccuracies were detected and corrected. Related edits have also been marked in yellow. Those edits are detailed below our responses to the editor comments.

Thank you very much for all your efforts! Please let us know if there is more input needed from our side.

Best regards,
Alexander, for the author team
* * *
***Executive Editor***

***Editor Comment (EC0):*** *While the release of MAGICC is clearly beyond your control, I think it ought at least to be possible to publish data and code such that someone who had a MAGICC licence could reproduce the work. For this a few things are needed, but you should be able to do them without too much difficulty.*

***Author Response (AR0):*** Thank you very much for raising these points! In order to address your concerns, we have moved and extended the published code, data, and model configuration information. Please see below for more details.

***Editor Comment (EC1):*** *Permanent archive of the exact version of your code. The GitLab URL has two issues: the first is that it's not permanent (the repository could be taken down tomorrow if the authors decided to), the second is that a GitLab URL does not indicate exactly which version of the code was used. The simplest way to achieve this is to upload a snapshot of the exact version of the code to an archival site, preferably one which issues DOIs. Zenodo is a good choice of archive, though others are available. You could either manually upload to Zenodo, or push a copy of your repository to GitHub and use the GitHub Zenodo integration.*

***Author Response (AR1):*** Done! The sea level model source code is now published on a Zenodo repository (https://doi.org/10.5281/zenodo.572395). The repository links to the supporting data which is available on a separate Zenodo repository (https://doi.org/10.5281/zenodo.572398). The final GMD DOI will be added to the source code repository once the paper is published. The model code and data availability section on p.18 has been changed accordingly.

***Editor Comment (EC2):*** *Permanent archive of the configuration and inputs used in the tests. This could be another Zenodo archive. Where you have used published CMIP input data, there is no need to duplicate this, so long as there is sufficient detail that another scientist could easily retrieve exactly the same input data. At the moment, a potential reproducer would be unable to ascertain which data they need to ask for.*

***Author Response (AR2):*** Done! Supporting input data and MAGICC model configuration information is now available on a dedicated Zenodo repository ([https://doi.org/10.5281/zenodo.572398](https://doi.org/10.5281/zenodo.572398)) which references the sea level model source code repository ([https://doi.org/10.5281/zenodo.572395](https://doi.org/10.5281/zenodo.572395)). We provide CMIP5 MAGICC input data as well as CMIP5 reference datasets for potential ocean temperature and thermal expansion for testing. The final GMD DOI will be added to the data repository once the paper is published. The calibration data used for the global glacier, Greenland, Antarctic, and land water storage sea level components have to be requested from the author teams of the corresponding studies. The model code and data availability section on p.18 has been revised accordingly.

***Editor Comment (EC3):*** *Document exactly the version of MAGICC used, so that a MAGICC licence holder could run the same version you ran.*

***Author Response (AR3):*** Done! On both Zenodo repositories, we clarify which model version has been used to produce the results presented in the manuscript (MAGICC version 7.0 beta). In addition, we now provide the MAGICC git hash to ensure the application of the exact same model version that has been used in this study.

We have edited the manuscript in response to the additional comments by referee #2. p.1 l.6 has been changed to "analysis", p.1 l.8 to "by", p.2 l.33 to "approaches", and p.3 l.3 to "do not change in the future". We have edited the sentence on p.11 l.28 to read: "Previous studies have shown that calibration methods for highly parameterized simple models do successfully show global convergence, even with a large number of free parameters." 2 has been changed to "two" on p.13 l.13. The statement on p.16 l.33 has been shortened to "It is also in line with observed past total sea level change (see Figure A.3). The close reproduction of selected reference data (Figure 3, Figure A.3), together with the consistent translation of climate forcing into a SLR response within the MAGICC model, and the comprehensive representation of relevant processes (e.g. the thermal expansion contribution produced by the CMIP5-consistent MAGICC ocean model and the inclusion of the land water storage sea level component) make the MAGICC sea level model a powerful addition to the existing sea level emulators." We have also changed p.17 l.29 to read "reference response of the calibration data." Figure 2 has been revised for improved readability.

We have detected and corrected rounding inaccuracies in Tables 6 and 7. The corresponding numbers have also been corrected in the abstract on p.1 and in the results section on p.15.

[revised manuscript text omitted]

---

## Author Response (AR3)

**GMD-2016-233 Author Response to Editor Review (Minor Revisions)**

Dear David,

Thank you very much for giving us the opportunity to transparently account for the parameter limits of the MAGICC model in Table 1. We are very sorry that the issue only got picked up at this late stage. Please accept our apologies for this oversight and the inconvenience caused!

Best regards,
Alexander, for the author team

*Executive Editor*

**Editor Comment (EC0):** *As discussed by email, you highlighted at the copy editing stage that one of the tables in the manuscript did not correctly account for the parameter limits imposed by the model. Since this involves a substantive change to the paper content, I am returning the manuscript for a minor revision to enable you to make this change, and to add a note explaining the parameter capping which occurs and citing the previous paper in which this is documented.*

**Author Response (AR0):** Thank you very much for allowing us to properly account for the MAGICC vertical diffusivity parameter limit that is relevant for the presentation of the calibration results in Table 1. The minimum vertical diffusivity $K_{z,min}$ of 0.1 cm²s⁻¹ is always effective in the model (as described in Meinshausen et al. 2011) and represents the lower bound for the $K_z$ calibration results presented in Table 1. Because $K_{z,min}$ is a separate parameter, the capping did not get picked up by the $K_z$ results presented in Table 1 for CESM1-CAM5, CMCC-CMS, GFDL-CM3, and GISS-E2-HCC. In order to be transparent to the reader, the corresponding $K_z$ entries should thus read 0.1000, as the model internally adjusts $K_z$ values lower than 0.1 to 0.1. Model results do not change, as running the model with, e.g., $K_z$ = 0.005 or $K_z$ = 0.1 leads to the identical outcome due to $K_{z,min}$. All calibrated parameter values as well as the GOFs are correct and do not change. Finally, we have included a sentence in the model calibration section that references and explains the role of the minimum vertical diffusivity $K_{z,min}$; p.12 l.3 now reads: "
[revised manuscript text omitted]